# Testing biological actions of medicinal plants from northern Vietnam on zebrafish embryos and larvae: Developmental, behavioral, and putative therapeutical effects

**My Hanh Tran[1,2], Thi Van Anh Nguyen[2], Hoang Giang Do[3], Trung Kien Kieu[4], Thi Kim Thanh Nguyen[5], Hong Diep Le[5], Gustavo Guerrero-Limon[1], Laura Massoz[6], Renaud Nivelle[1], Jérémie Zappia[1], Hai The Pham[2,4], Lai Thanh Nguyen[4], Marc Muller[1]***

1 Laboratory for Organogenesis and Regeneration, GIGA I3, Université de Liège, Liège, Belgium,
2 Department of Microbiology, Vietnam National University of Science, Faculty of Biology, Hanoi, Vietnam,
3 Center for Research and Technology Transfer, Vietnam Academy of Science and Technology, Hanoi, Vietnam, 4 GREENLAB, Center for Life Science Research (CELIFE), Vietnam National University of Science, Faculty of Biology, Hanoi, Vietnam, 5 Department of Plant Science, Vietnam National University of Science, Faculty of Biology, Hanoi, Vietnam, 6 Zebrafish Development and Disease Model laboratory, GIGA Stem cells, Université de Liège, Liège, Belgium

* m.muller@uliege.be

**Data Availability Statement:** Data relevant to this study are available from NCBI's Gene Expression

## Abstract

Evaluating the risks and benefits of using traditional medicinal plants is of utmost importance for a huge fraction of the human population, in particular in Northern Vietnam. Zebrafish are increasingly used as a simple vertebrate model for testing toxic and physiological effects of compounds, especially on development. Here, we tested 12 ethanolic extracts from popular medicinal plants collected in northern Vietnam for their effects on zebrafish survival and development during the first 4 days after fertilization. We characterized more in detail their effects on epiboly, hatching, growth, necrosis, body curvature, angiogenesis, skeletal development and mostly increased movement behavior. Finally, we confirm the effect on epiboly caused by the *Mahonia bealei* extract by staining the actin filaments and performing whole genome gene expression analysis. Further, we show that this extract also inhibits cell migration of mouse embryo fibroblasts. Finally, we analyzed the chemical composition of the *Mahonia bealei* extract and test the effects of its major components. In conclusion, we show that traditional medicinal plant extracts are able to affect zebrafish early life stage development to various degrees. In addition, we show that an extract causing delay in epiboly also inhibits mammalian cell migration, suggesting that this effect may serve as a preliminary test for identifying extracts that inhibit cancer metastasis.

## Introduction

In recent years, the pharmacological study of traditional medicine knowledge in plant research received considerable interest. Medicinal plants contain many phytochemicals such as

Omnibus at GEO Series accession number GSE207770.

**Funding:** This research was funded by ARES (Académie de Recherche et d'Enseignement Supérieur)(https://www.ares-ac.be/fr/), grant number PRD17: "Exploring the medical, (eco)-toxicological and socio-economic potential of natural extracts from Northern Vietnam." M.H.T. was a fellow from ARES, G.G-L. had a fellowship from EU project MSCA_ITN PROTECTED and M.M. is a "Maître de Recherche" at FNRS. The funders had no role in study design, data collection and analysis, decision to publish, or preparation of the manuscript.

**Competing interests:** The authors have declared that no competing interests exist.

vitamins, carotenoids, terpenoids, flavonoids, polyphenols, alkaloids, tannins, saponins, enzymes, and minerals that present biological activities on other organisms. They have antimicrobial activities, anti-inflammatory [1, 2], anti-hyperuricemic [3], antioxidant [4, 5] and anticancer properties [6]. Therefore, these phytochemicals have been utilized to prevent or cure a variety of disorders [7, 8].

Southeast Asia has one of the highest levels of biodiversity on the planet due to its complicated geological history. Geologic movements in the past, along with a tropical environment, aided speciation and formed in this region one of the most diverse biotas on earth. As a Southeast Asian country, Vietnam also possesses a diverse biota. In addition, with its land spanning a wide range of longitudes and various geological features, the natural environment in Vietnam provides a rich source of unusual and potentially intriguing biological components, many of which may be native to Vietnam and hence have not yet been documented elsewhere. On the scientific aspect, at present, the medicinal potential of this biological diversity is largely unexplored in Northern Vietnam. In this region, traditional medicine is mainly used locally for prevention and treatment of high frequency diseases, such as: infection, inflammation, immunological (including autoimmune) and heart disease, metabolic and neurologic disorders. Only some systematic reviews have been published of materials used in Vietnamese traditional medical practice [9, 10], and in Vietnamese [11–14]. Traditional medicine mostly relies on experience passed on from one generation to the next to use specific plant extracts for certain pathologies. However, this practice suffers from the absence of systematic, interdisciplinary, large-scale, and comprehensive studies on the biological and pharmacological properties of the medicinal plants used. Furthermore, medicinal plants are often seen as "natural" products that may be administered for extended periods, occasionally resulting in severe adverse health outcomes. Further evidence-based studies and meta-analyses are required to provide a more objective assessment.

Zebrafish is commonly used as a vertebrate model organism [15] initially demonstrating its use in embryonic developmental biology but also increasingly in toxicology, pharmacology, and vertebrate behavioral biology in recent years [16–18]. The zebrafish offers numerous benefits over traditional animal models (rabbit, rat, and mice) [2, 19]. First, although being a complete and independent life form, the zebrafish embryo up to the free-feeding stage at 5 days post fertilization is generally considered as larval stage; hence it can be used without raising ethical issues. Second, zebrafish are easy to maintain, and embryo yields are tremendous (50–300 embryos/pair/week). Third, the developmental process of a zebrafish embryo represents that of other vertebrates with highly similar embryogenesis and organogenesis, but much faster, which can be observed clearly through the transparent embryos [20]. Moreover, being one of the first sequenced and annotated genomes, the well-known zebrafish genome shares numerous similarities with higher vertebrates, including humans [21], especially in structure and function of certain CYP family genes involved in drug metabolism [22]. When these factors are considered together, zebrafish is a highly potent model for testing drug's preclinical (developmental) toxicity today.

The "Zebrafish Embryo Toxicity" (ZET) test is included as an alternative to animal testing [23] and can be performed with a variety of endpoints [24–26], which would be any "...biological or chemical process, response, or effect, as measured by a test." [27]. Within this scope, in this study, twelve medicinal plants from northern Vietnam were selected, based on traditional use as well as socio-economic issues, for a rapid screening test using the zebrafish model. Focusing on the first 5 days of development, we utilize this approach to evaluate effects on early embryogenesis, of particular interest when treating pregnant women, but we also rapidly assess many pathways and processes that may play a role in pathogenesis at later life stages of crude extracts obtained from the six most promising plants. Finally, the effect on epiboly of

the *Mahonia bealei* extract was further investigated and an effect on mammalian cell migration was confirmed. First tests investigating major purified chemical components of this extract were also performed.

## Materials and methods

### Plant material

Medicinal plants (Table 1) were selected based on the recipes of traditional medicines and collected from the North-Western mountainous areas of Vietnam in the dry season of 2019. They were identified, coded (with Vietnamese and Scientific names) and placed at the Museum of Biology, Faculty of Biology, VNU University of Science, Vietnam National University, Hanoi, Vietnam.

### Preparation of plant extracts

Dried plant powders were used for extraction with ethanol solvent. Each dried powder sample was suspended with solvent at the ratio of 1:10 (w/v) and placed into an ultrasonic incubator (S100H; Elma GmbH, Singen, Germany) for 30 min at 45°C. The mixture was shaken at 200 rpm for 60 min at room temperature in an orbital incubator shaker (Gyromax™ 747 Incubator Shaker; Amerex Instruments, Concord, CA USA). The supernatant was separated from the plant residues by centrifugation (Avanti J-E; Beckman Coulter, Brea, CA USA) at 10000 rpm for 20 min at 10°C, then filtered through Whatman No. 1 filter paper; the obtained filtrate was dried completely in a vacuum rotary evaporator (HS-2005S-N, Hahnshin, Gimpo, South Korea). The dried extracts were stored at 4°C. Just before experimentation, the extracts were dissolved in dimethyl sulfoxide (DMSO) to obtain stock solutions, which were then diluted to different concentrations for further analyses.

### Zebrafish maintenance

Animal care and all experimentation were conducted in compliance with Belgian and European laws (Authorization: LA1610002). All experiments and the entire study were evaluated

**Table 1. Medicinal plants used for toxicity screening.**

| Label/ Code | Scientific names (Vietnamese name) | Voucher number | Sampling: GPS coordinates, province, date collected | Parts of interest |
|---|---|---|---|---|
| KT02 | *Lactuca indica* L. (Bo cong anh cao) | HNU 024108 | 21°00'57.3"N 107°19'02.6"E (Lạng Giang, Lạng Sơn); spring 2019 | Fresh stem |
| KT06 | *Anisomeles indica* (L.) Kuntze (Phong phong thao) | HNU 024777 | 22°26'18.7"N 103°55'25.9"E (Bat Xat, Lao Cai); fall 2017 | Dry stem/ leaves |
| KT07 | *Clerodendrum cyrtophyllum* Turcz. (Bo may) | HNU 024106 | 22°26'06.7"N 104°57'55.1"E (Bac Quang, Ha Giang); fall 2017 | Dry leaves |
| KT08 | *Pericampylus glaucus* (Lam.) Merr. (Tiet de la day) | HNU 024778 | 22°26'18.7"N 103°55'25.9"E (Bat Xat, Lao Cai); fall 2017 | Dry leaves |
| KT09 | *Mahonia bealei* (Fortune) Pynaert (Hoang lien) | HNU 024779 | 22°34'01.5"N 103°46'47.9"E (Bat Xat, Lao Cai); fall 2017 | Dry stem |
| KT11 | *Gnetum montanum* Markgr. (Day gam nui) | HNU 024781 | 22°34'47.0"N 105°06'39.8"E (Vi Xuyen, Ha Giang); summer 2017 | Dry stem |
| KT12 | *Tacca chantrieri* André (Rau hum hoa tia) | HNU 024782 | 22°34'47.0"N 105°06'39.8"E (Vi Xuyen, Ha Giang); fall 2018 | Dry stem |
| KT14 | *Mallotus barbatus* Müll.Arg. (Bum bup) | HNU 024784 | 20°32'02.6"N 105°23'55.8"E (Lac Son, Hoa Binh); summer 2017 | Dry root |
| KT15 | *Aganope balansae* (Gagnep.) P.K.Loc (Man man) | HNU 024785 | 22°26'06.7"N 104°57'55.1"E (Bắc Quang, Hà Giang); fall 2018 | Dry stem |
| KT17 | *Stixis scandens* Lour. (Trung cuoc) | HNU 024787 | 22°26'06.7"N 104°57'55.1"E (Bắc Quang, Hà Giang); fall 2019 | Dry leaves |
| KT19 | *Croton kongensis* Gagnep. (Kho sam) | HNU 024789 | 20°32'02.6"N 105°23'55.8"E (Lac Son, Hoa Binh); fall 2017 | Dry leaves |
| KT20 | *Tinospora sinensis* (Lour.) Merr. (Day dau xuong) | HNU 024790 | 20°32'02.6"N 105°23'55.8"E (Lạc Sơn, Hòa Bình); fall 2018 | Fresh stem |

by the Ethical Committee of the Université de Liège, Belgium and the study protocols were approved under the file numbers 1076 and 13–1506. Zebrafish (*Danio rerio*) of the AB strain and the *Tg(fli1:EGFP)* [28] transgenic line were obtained from breeding facilities at the GIGA-Institute, Liege, Belgium. The water characteristics were as follows: pH = 7.4 and temperature = 28˚C. The lighting was 14/10-h light/dark. Embryos were used and staged as described [29, 30]. Fish were mated and spawning was stimulated by the onset of light. The day before breeding, adult male and female zebrafish were set up in several breeding tanks, separated by a clear plastic wall. After the light was turned on the next morning, walls were removed, natural mating occured and the eggs were collected between 30 minutes to 2 hours after spawning. After sorting, fertilized clean eggs are moved to Petri dishes and incubated at 28˚C in E3 medium (5 mM Na Cl, 0.17 mM KCl, 0.33 mM CaCl2, 0.33 mM MgSO4, 0.00001% Methylene Blue). The age of the embryos and larvae is indicated as hours post fertilization (hpf) [31].

## Medicinal plant extract exposure

The plant extract solutions at different concentrations for toxicity tests were obtained by dilution of the stock solutions in embryo medium (E3). Treatments were administered starting at 2 hpf to zebrafish embryos distributed in pools of 25 individuals into 6-well plates in E3 medium until 120 hpf. The larvae were rinsed every 24 hours in E3 until 96 hpf and the treatments were renewed each day using a freshly prepared plant extract solution to maintain oxygen and nominal concentrations constant during the assay and to remove fungi or other organisms that could develop in the dish [32]. Untreated control batches received only the E3 with 0.5% DMSO used for the stock solutions. Three independent biological replicates were run for each concentration of plant extracts and controls. The ranges of concentrations were selected based on the results of the preliminary test which identified a concentration to induce death and morphological defects. The larvae were rinsed twice in E3 before observation, survival and morphological/ developmental defects were assessed using a Leica M165 FC stereomicroscope (Leica Microsystems©, Wetzlar, Germany). Developmental morphological abnormalities were recorded as the numbers of larvae presenting at least one morphological defect by using parameters such as retardation, spinal curvature, yolk sac edema, delay in pigmentation and hatching delay [33]. Experiments having more than 5% abnormal development in the controls were discarded. Each treatment group was analyzed without preconception, especially regarding the specific defect observed.

## Angiogenic effects analysis

To assess the anti-angiogenic potential of the plant extracts, we used the transgenic *Tg(fli1a: EGFP)* zebrafish that continuously express the green fluorescent protein EGFP under control of the *fli1* promoter in the endothelial cells (which make blood vessels) [28]. The embryos were exposed to sub-$LC_{10}$ values of plant extracts, to check the activity of medicinal plants on angiogenesis. After 24 hpf, the embryos were incubated in a pronase solution (2mg/ml) at room temperature typically for 3–5 minutes. For mechanical dechorionation, the embryo suspension was pipetted back and forth using a Pasteur pipet [34] before thorough washing (3 or 4 times) in E3 medium to remove pronase. The released embryos were then exposed to the plant extracts and examined at 72 hpf [35] using a Leica M165FC stereomicroscope for the presence and the number of ectopic vessels in the sub-intestinal vessel (SIV) region as an indication of pro-angiogenesis effects [36]. The mean sprout number was assessed by the sum of the number of sprouts present in each sample group over the total number of embryos in the sample group. Sprout intensity of the sample group was rated as "+", "++", and "+++",

representing the percentage of embryos with one, two, and three or more sprouts in the SIV regions, respectively. Images were processed using the ZEN software (Zeiss, Jena, Germany).

## Cartilage staining

Alcian blue staining was used for optimal visualization of cartilage structures in the head of the embryos (head malformations) as described in previous studies [37, 38]. At 120 hpf, the larvae were sacrificed with Tricaine. The exposure medium was then removed, and the larvae were fixed with 4% paraformaldehyde in PBS solution (4% PFA-PBS) at 4˚C overnight. The larvae were rinsed 3 times with PBST during 10 mins, then were stained with 0.04% Alcian blue in 80% ethanol with 20 mM MgCl2 overnight. Background of the staining was removed by washing with 80% ethanol/20 mM MgCl2/water several times. After that, bleaching was conducted by adding 1 ml of H2O2 3%/KOH 0.5% during 20–60 mins. The larvae were then rinsed with graded concentrations of glycerol in KOH: 25% glycerol/0.1% KOH and 50% glycerol/0.1% KOH and kept in this solution for storage [39]. The cartilage-stained larvae were transferred and embedded on slides in the correct orientation (ventral-up position for pharyngeal cartilage and dorsal-up position for other craniofacial elements) with absolute glycerol and photographed using a digital camera mounted on an Olympus SZX10 stereomicroscope (Olympus, Antwerp, Belgium).

## Behavior analysis

Behavioral tests were conducted on zebrafish larvae at $> 98 \sim 120$ hpf and every test was performed between 10:00 to 13:00 to maintain a constant position in the circadian cycle. Prior to each behavioral test, the zebrafish larvae were inspected under a stereomicroscope to select and transfer to the testing plates only individuals devoid of any malformation that might interfere with mobility outcome (*e.g.* yolk sac or pericardial edemas, spinal aberrations, aberrations in pigmentation, and/or loss of equilibrium, *etc*). The treated larvae were transferred to 96-well microplates with one embryo in each well containing 100 μl of E3 medium (24 embryos per group). Movements of exposed and control individual larvae were recorded using a ViewPoint® Zebrabox system and its tracking software (ViewPoint Life Sciences, Lyon, France). The light level was set to 20% on the ViewPoint software (7.45 Klux, TES 1337 light meter), while infrared light (850 nm) was used to track larval activity. We used an assay with alternating, 10 min light and 10 min dark cycles to study the dynamics of the photo-motor response in different light and dark phases during the experiment. The locomotion measurements were assessed for 1 hour by recording free swimming activity, swimming distances and speeds. In the first 20 min., the larvae were kept in the light for habituation, while the recordings were discarded from the analysis, then they were subjected to two cycles of alternating 10 min. dark and light periods to record their swimming behavior. The video and tracking software was set to screen larval locomotion behavior in 30 seconds intervals, the "distance travelled" and the "time spent active" were determined and, from these parameters, the mean swimming speed was also calculated by dividing the cumulated distance travelled by the total time spent active.

## Cell culture and *in vitro* wound healing

Mouse Embryonic Fibroblasts (MEFs) (STO: CRL-1503) were grown in DMEM-HG (D-7777, Sigma) supplemented with 10% FBS (v/v) and 1% antibiotics (50 U/ml penicillin– 50 ug/ml streptomycin). A cell viability/proliferation test based on the detection of ATP produced by viable cells (CellTiter-Glo™ Luminescent Cell Viability Assay Kit; Promega) was performed using

the limit test to find the maximum concentration having no effect (the NOEC) and the lowest concentration having an effect (LOEC).

The *in vitro* scratch assay was performed on MEF cells to examine wound-healing capacity as previously reported [40, 41]. MEFs were seeded into 24 well plates and grown until reaching 80% confluence. Scratches (0,6 ± 0,05mm) were made using SPL ScarTM Scratcher (SPL Life Sciences, Pocheon, South Korea) and cells were washed with PBS twice to ensure there were no cells within the scratch area. As the extracts were dissolved in DMSO, DMSO 0.1% was added to the medium DMEM-HG + 10% FBS (v/v) + 1% antibiotics (50 U/ml penicillin– 50 ug/ml streptomycin) with β-Mercaptoethanol as the negative control. Scratches were photographed under the microscope immediately after scratching and after 12h, 14h, 18h and 22 h (until the scratches in the control wells were restored to about 100%) incubation at 37˚C in a 5% CO2 incubator to document the wound-healing process. The images were processed using the Macro "MRI Wound Healing Tool" of ImageJ software to determine the areas of the wounds at the marked location over various time frames to evaluate the influence of the extract on the cellular wound healing capacity. We used the following formula:

$$\text{Wound area recovery rate (\%)} = \left(1 - \frac{\text{scratch area at specific time}}{\text{scratch area at 0th hour}}\right) \times 100\%$$

## F-actin staining

Wildtype embryos at 8 hpf (70% epiboly stage) were fixed in 4% paraformaldehyde (PFA) in PBS at 4˚C overnight, and then washed 3 times with 0.1% Tween-20 PBS for 5 min each. After washing, unspecific sites were blocked with 4% BSA in PBS for 1 hour at 4˚C, then rinsed with PBS. Afterward, the embryos were incubated with phalloidin-Atto 532, a high-affinity probe for F-actin (Merck, Darmstadt, Germany) for 1 hour at 4˚C, then washed 3 times for 5 min each time with PBS. Subsequently, they were mounted on Vectashield mounting media (Prolong Gold antifade®, Invitrogen, Waltham, MA USA), imaged under a Leica SP5 confocal microscope (Leica, Wetzlar, Germany), and processed using LSM software. The figures were assembled using Image J (NIH, Bethesda, MD USA).

## RNA extraction

RNA was extracted from pools of 75 larvae using the RNA mini extraction kit (Qiagen, Hilden, Germany). We chose to use whole larvae, including the yolk, as we intended to determine the entire RNA pool for evaluation of transcription, but also maturation and degradation. Samples were lysed in RLT+ buffer with β-mercaptoethanol (Sigma-Aldrich, Overijse, Belgium) and homogenized 10 times with a 26-gauge needle in a 1 ml syringe. 22 μl of RNAse free water were used to resuspend total RNA. RNA extract was treated with DNAseI (Qiagen, Hilden, Germany) to avoid DNA contamination. The quantity (ng/μl) and quality of each extract was assessed by nanodrop spectrophotometer measurements (Thermofisher Scientific, Waltham, MA USA). Poor quality (260/280 < 2; 260/230 < 2) samples were subsequently purified by Lithium chloride precipitation, followed by 2 times pellet washing with 70% ethanol and resuspended in 40 μl of RNAse-free water and stored at -80˚C. The integrity of total RNA extracts was assessed using a BioAnalyzer (Agilent, Santa Clara, CA USA) providing RIN (RNA Integrity Number) scores for each sample.

## RNA sequencing

cDNA libraries were generated from 100 to 500 ng of total RNA using the Illumina Truseq mRNA stranded kit (Illumina, San Diego, CA USA) according to the manufacturer's

instructions. cDNA libraries were then sequenced on a NovaSeq sequencing system, in 1 x 100 bp (single-end). Approximatively 20-25M reads were sequenced per sample. The sequencing reads were processed through the Nf-core rnaseq pipeline 3.0 [42] with default parameters, using the zebrafish reference genome (GRCz11) and the annotation set from Ensembl release 103 (www.ensembl.org). Differential gene expression analysis was performed using the DESeq2 pipeline [43]. Pathway and biological function enrichment analyses was performed using the WEB-based "GEne SeT AnaLysis Toolkit" (http://www.webgestalt.org) based on the integrated GO (Gene Ontology), KEGG (Kyoto Encyclopedia of Genes and Genomes) [44, 45], Panther and WikiPathways databases. An additional database was constructed using the Gene-mutant/Phenotype database from zfin (zfin.org).

## Identification of alkaloids from *M. bealei* ethanolic extract

The alkaloid components from *M. bealei* ethanolic crude extract were identified by the HPLC-DAD method. 9 reference compounds, including jatrorrhizine, isotetradine, 3-hydroxy-8-oxopalmatine, 8-oxypalmatine, oxyberberine, rugosinone, 8-oxyberberubine, berberine, palmatine, were collected from the in-house library of the "Center for Research and Technology Transfer, Vietnam Academy of Science and Technology". HPLC-grade solvents were purchased from Scharlau (Barcelona, Spain) and Merck Millipore (Darmstadt, Germany). HPLC-DAD data were generated on a Thermo Ultimate 3000 system, consisting of a quaternary pump, an autosampler, a column oven, and a diode array detector. A Hypersil GOLD HPLC column (250 mm x 4.6 mm, 5 µm) was used at 40˚C. The solvent conditions were optimized to detect as many compounds as possible, specifically potassium phosphate buffer (10 mM, pH 5.0) and acetonitrile (0.1% of acetic acid) were set as solvent channels A and B, respectively. A flow rate of 1.0 mL per min was selected with a linear gradient from 20% to 80% B over 20 min, followed by 5 min washing with 80% B and 5 min of the initial condition. The injection volume for each sample was 5 µL. The UV detector was set at 270 nm for the simultaneous determination of 9 standard compounds. The compounds were detected in the extract by comparing their retention times and UV spectra to those of the reference compounds which were analyzed under the same condition. Berberine and palmatine were purified as previously described [46].

## Data and statistical analysis

All experiments were performed at least three times, mostly on different dates but always on different clutches from different parents. Statistical analyses were performed using GraphPad Prism for Windows (version 8.0.1) using the appropriate tests as indicated. $LC_{50}$, $LC_{10}$, $EC_{50}$ and $EC_{10}$ were calculated by plotting the surviving/affected larvae against the log transformed tested concentration and the obtained curve was fitted to a sigmoidal concentration–response relation according to the following equation:

$$y = \frac{Bottom + (Top - Bottom)}{1 + 10 \log EC50 - x}$$

where bottom and top represent, respectively, the lowest and the highest y-value (% survivors/ affected). The resulting calculated log EC/LC were extracted, and their corresponding 95% Confidence Intervals (C.I.) are given.

Raw behavioral data sets consisted of tables holding the positions of each larva in each video frame (30 frames/second). This table was first cleaned to eliminate very short, oscillating and likely artefactual movements, and then aggregated into 30-second periods for further analysis. These data were transferred to R version 4.0.2 [47] to analyze motility during the dark

and light phases. To assess behavior, we used linear mixed effect (LME) models within the "nlme" package. Three dependent variables were used, either the "mean time spent active" (seconds), the "mean distance travelled" (mm), or the "mean swimming speed" (calculated as the mean distance travelled/mean time spent active) within each 30 second period, with "compound" as the categorical independent variable, and batch as a random effect. The "Anova" command within the "car" library was used to extract the results for the main effects whereas the "lsmeans" command within the "emmeans" library was used as a post-hoc test to compare groups against one another while adjusting for the means of other factors within the model [48]. Type II sum of squares was used for the model. Significance was assigned at $p = < 0.05$.

## Results

### Effects of medicinal plant extracts on survival and malformation of zebrafish embryos

Ethanol extracts from 12 medicinal plants collected from Northern Vietnam were examined according to the zebrafish embryo toxicity test [49]. To assess the survival and developmental defects caused in the zebrafish embryos/larvae by exposure to plant extracts, embryos were treated with a range of concentrations based on a preliminary, range-finding test for each extract starting at 2 hpf and assessed every day until 96 hpf for lethality and occurrence of morphological defects (Table 2).

For all the tested plants, we were able to define lethal concentrations, with only KT12 presenting an $LC_{50}$ below 100μg/mL which would classify it as toxic [49], while KT17 was the least toxic extract ($LC_{50} > 1.3$ mg/mL at all stages). In terms of developmental defects, KT17 stood out again with the highest $EC_{50}$ for any observed defect, while KT09 and KT07 presented the lowest $EC_{50}$ at 72 and 96hpf. These three extracts also showed the highest teratogenic indices (TI), either due to the low lethal effect (KT17) or to the strong effect on embryogenesis. KT12 extract caused no developmental defects, due to its high level of toxicity. Based on these screening results, we decided to focus on a group of plants that all caused developmental effects but covering a range of $EC_{50}$ and $LC_{50}$ values (KT02, KT09, KT11, KT14, KT15, KT20), to

**Table 2. Calculated toxicological indices for all 12 studied extracts.**

| Code of extract | LC$_{50}$ (μg/mL) | | | | EC$_{50}$ (μg/mL) | | | | TI |
|---|---|---|---|---|---|---|---|---|---|
| | 24h | 48h | 72h | 96h | 24h | 48h | 72h | 96h | 96h |
| **KT02** | >2000 | >2000 | 437–508 | 246–255 | 547–724 | 216–234 | 202–214 | 203–209 | 1.22 |
| KT06 | 248–420 | 181–270 | 175–271 | 171–262 | No defects | | 105–215 | 125–279 | 1.14 |
| KT07 | 599–770 | 430–738 | 335–444 | 236–294 | 273–1411 | 132–235 | 94–101 | 48–71 | 4.59 |
| KT08 | 466–682 | 471–613 | 390–604 | 460–571 | No defects | | 206–685 | 218–391 | 1.73 |
| **KT09** | 243–313 | 226–277 | 181–191 | 128–132 | 173–198 | 106–147 | 53–75 | 56–63 | 2.20 |
| **KT11** | 262–296 | 250–278 | 232–257 | 138–144 | 270–314 | 173–186 | 97–123 | 77–83 | 1.76 |
| KT12 | 94–104 | 79–107 | 70–91 | 68–91 | No defects | | | | |
| **KT14** | >2000 | >2000 | 267–1202 | 178–192 | 502–519 | 236–253 | 253–316 | 228–257 | 0.76 |
| **KT15** | 250–267 | 227–244 | 202–224 | 164–169 | 295 | 180–192 | 114–124 | 98–113 | 1.57 |
| KT17 | 1631–2082 | 1553–1779 | 1470–1691 | 1309–1413 | No defects | 844–1113 | 707–915 | 555–827 | 2.03 |
| KT19 | 148–274 | 142–274 | 105–189 | 103–161 | >2000 | >2000 | 155 | 88–146 | 1.08 |
| **KT20** | 552–629 | 373–425 | 365–418 | 365–418 | 540–682 | 322–452 | 304–340 | 271–403 | 1.17 |

The calculated log EC/LC are shown as their corresponding 95% Confidence Intervals (C.I.) are given. Rare or irreproducible phenotypes were ignored. Teratogenic Index (TI) at 96 hpf highlighted in grey, selected extracts in bold.

determine the specific defects they cause, starting at early stages and at concentrations around $LC_0$, $LC_{10}$ and $LC_{25}$ for each extract (Fig 2).

While no effect on survival or malformation was observed at any time point following exposure to 0.5% DMSO, the highest concentration of solvent used in these experiments, exposure of zebrafish embryos to medicinal plant extracts resulted in decreased survival depending on the concentration of the specific extract. Survival rates were analyzed for several concentrations and $LC_{50}$ was determined for each extract (Table 2). The lethal toxicity was gradually increased from KT20, KT02, KT14, KT15, KT11, to KT09 extracts, based on $LC_{50}$-96h, as summarized in Table 2 and Fig 1. The results of lethal concentrations clearly show that all the compounds are lethal to zebrafish larvae at various concentrations. With the possibility to observe early developmental effects during the first 4 days, we concentrated on perturbations such as dorsal curvature, yolk sac edema, hemorrhage, delayed hatching, or altered size. The fraction of the surviving embryos presenting any of the above-described morphological abnormalities was determined for the different concentrations of each extract (Fig 1). The teratogenic Index (TI) steadily increased from KT14, KT20, KT02, KT15, KT11, to KT09 extracts (Table 2). All extracts have TIs greater than 1, with the exception of KT14, indicating that all of these chemicals have the potential to cause abnormalities in zebrafish embryos. The TI of KT14 is less than 1, suggesting that this is a milder teratogenic agent for fish embryos and typically surviving embryos/larvae would not have deformities.

## Morphological effects of six medicinal plant extracts on zebrafish embryos

Representative images are given for the various concentrations of each extract (Fig 2). In general, the effects of all extracts are dose-dependent (Fig 2 and Table 3): not only did the proportion of dead and malformed embryos increase with increasing concentration, but so did the severity and complexity of phenotypes–as evidenced by the number of phenotypes and the fraction of mixed or multiple phenotypes (Table 3), particularly in severely deformed embryos. Furthermore, the severity of specific traits increased (*e.g.* bigger edemas, more strongly curved tail, or darker yolk necrosis) (Table 3). Embryos and larvae exposed to extracts at high concentrations, near the $LC_{50}$ had significant systemic morphological abnormalities.

A noteworthy feature observed was that all the extracts generated dumbbell-shaped yolk embryos at the stage of 8–10 hpf, as best illustrated by the KT09 extract which had the highest incidence (Fig 2B and Table 3). It began with the presence of a dumbbell-shaped yolk between 8 and 10 hpf, followed by a clear delay in development at 1dpf that increased with increasing extract concentration. At 2 and 3dpf, the most striking feature was a delay in hatching, while in 3 and 4dpf larvae the lower concentrations of treatment induced some local malformations such as yolk sac edema and curvature in the surviving larvae. The survival rate of embryos declined strongly within the first 24 hours (S1 Fig). We also find the presence of a dumbbell-shaped yolk and hatching retardation when embryos are exposed to a high concentration of KT11 extract. In addition, the KT11 extract caused an even stronger delay in hatching; the embryos did not hatch after 4 days, and some of them died inside the chorion. The survival rate of embryos was reduced at 24hpf, constant until 72hpf, and reduced again at 96hpf (S1 Fig), implying that this extract causes death at an early stage (before 24hpf) and again later, possibly due to the hatching delay. RNA-Seq analysis at different time points would help to understand the mechanisms involved. The presence of dumbbell-shaped yolk after KT14 extract treatment was observed at 200 μg/ml and 10hpf, however it appears to be recovered at later stages, as no hatching or developmental delay was seen at 24 or 48hpf. Intriguingly, lethality of this extract increased dramatically at 72 hpf (S1 Fig), after hatching, implying that the chorion plays a protective role against its toxicity. At high concentrations, the KT15 extract

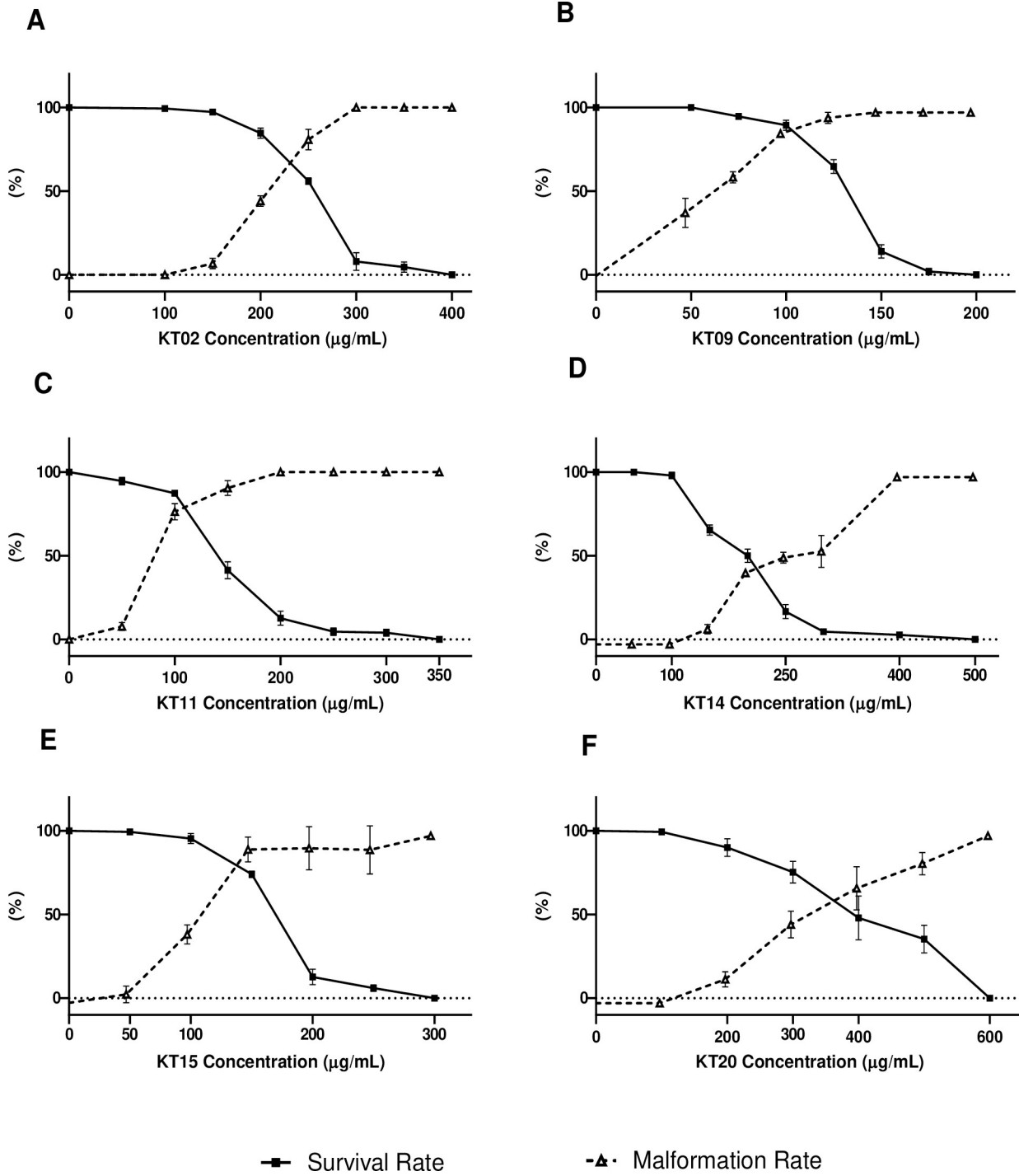

**Fig 1. Survival and morphological defects of larvae after exposure to medicinal plant extracts.** Fraction of surviving larvae or larvae presenting any morphological defect at 96 hpf upon exposure to KT02 (A), KT09 (B), KT11 (C), KT14 (D), KT15 (E) and KT20 (F). Experiments were performed at least in triplicate with each time n = 50 larvae per point. Fraction of larvae presenting any morphological defects are represented in dotted lines.

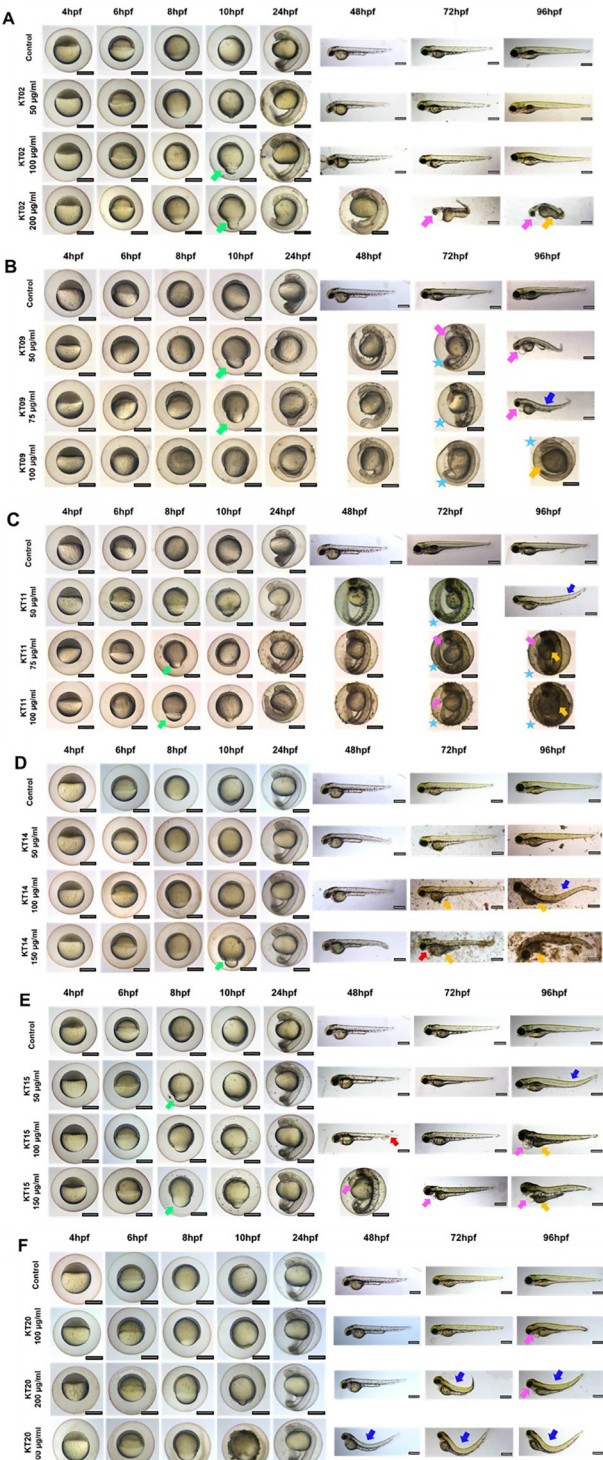

**Fig 2. Morphological defects at different time points of exposure.** A-F: Representative images of 4–96 hpf larvae exposure to 6 plant extracts at the indicated concentrations. The morphology of embryos was observed and captured under a stereomicroscope. Scale bars = 400 μm. Green arrows indicate dumb-bell shape yolk, blue arrows indicate dorsal curvature, pink arrows indicate yolk sac edema, red arrows indicate hemorrhage orange arrows indicated necrosis, and stars indicate delayed hatching.

**Table 3. Frequencies of specific defects caused by 6 plant extracts (numbers were recorded at the concentrations of $LC_{10}$ and $LC_{25}$).**

| Defects | Control | KT02 | | KT09 | | KT11 | | KT14 | | KT15 | | KT20 | |
|---|---|---|---|---|---|---|---|---|---|---|---|---|---|
| | | $LC_{10}$ | $LC_{25}$ | $LC_{10}$ | $LC_{25}$ | $LC_{10}$ | $LC_{25}$ | $LC_{10}$ | $LC_{25}$ | $LC_{10}$ | $LC_{25}$ | $LC_{10}$ | $LC_{25}$ |
| Dumb-bell shaped yolk (10hpf) | 3/150 | 17/150 | 47/150 | 31/149 | 65/147 | 19/149 | 43/147 | 17/150 | 44/149 | 18/150 | 50/149 | 11/150 | 32/149 |
| Oedema (>24 hpf) | 2/147 | 9/137 | 55/115 | 18/141 | 55/117 | 23/138 | 50/116 | 9/135 | 21/112 | 29/141 | 51/113 | 11/141 | 29/120 |
| Necrosis (72/96hpf) | 0/147 | 2/137 | 17/115 | 5/141 | 52/117 | 17/138 | 50/116 | 27/135 | 45/112 | 16/141 | 48/113 | 5/141 | 15/120 |
| Hemorrhage (72/96hpf) | 0/147 | 0/137 | 5/115 | 3/141 | 9/117 | 0/138 | 5/116 | 3/135 | 7/112 | 0/141 | 1/113 | 0/141 | 0/120 |
| Curvature (72/96 hpf) | 1/147 | 5/137 | 19/115 | 7/141 | 20/117 | 3/138 | 8/116 | 10/135 | 31/112 | 9/141 | 30/113 | 19/141 | 41/120 |
| Hatching delay (72/96 dpf) | 2/147 | 0/137 | 24/115 | 31/141 | 56/117 | 41/138 | 62/116 | 0/135 | 4/112 | 2/141 | 5/113 | 0/141 | 0/120 |

caused developmental delay and after three days of exposure, this extract was highly toxic to embryos. The treated embryos displayed substantial defects, such as edema, altered size, and necrosis, which could lead to mortality in a later stage. This finding explained why the survival rate of embryos was lower after 72 hours of treatment. KT02 and KT20 extracts appear to be safer and produce fewer abnormalities than the other extracts in the group; nevertheless, some defects were noticed at high concentrations of these extracts. Embryos exposed to KT02 had a lower survival rate after 48 hours, and embryos exposed to KT20 had a lower survival rate after 24 hours (S1 Fig).

## Effects of six medicinal plant extracts on angiogenesis in zebrafish larvae

Angiogenesis plays an important role in a wide range of physiological processes such as wound healing and fetal development. In fact, many diseases are associated with an imbalance in the regulation of angiogenesis in which there is either excessive or insufficient blood vessel formation. The fact that some extracts caused a hemostasis phenotype in larvae prompted us to investigate their anti-angiogenic properties. To that end, the zebrafish transgenic line *Tg (fli1:EGFP)* (y1Tg) displaying fluorescent endothelial cells was used to visualize blood vessels and detect effects on angiogenesis. In zebrafish, a basal blood circulatory loop is set up at 24hpf, then angiogenic vessel development extends the system, including the sub intestinal vessels (SIVs) at 72hpf. We thus investigated the effect of six plant extracts on changes in SIV development at 72hpf.

Six plant extracts were shown to promote changes in SIV formation to various degrees, at exposure concentrations substantially lower than the corresponding $LC_{10}$. The most striking feature we observed was the formation of longer, more numerous, and ectopic sprouts projecting outwards from the sub intestinal vessel basket (ramification). We classified the individuals according to their SIV formation from WT (-) through increasing levels of ectopic sprouts (+, ++, +++,...) (Fig 3B) to formation of hemorrhagic patches (Fig 3A). Looking at the frequency of embryos with varying degrees of ectopic sprouts in the SIVs (percentage of abnormal phenotypes) when exposed to increasing concentrations of each plant extract, it appears that the treatment with KT11 and KT14 trigger the most ectopic angiogenesis of SIVs compared to controls, followed by exposure to KT09 and KT02. These defects worsened in a dose-dependent manner. On the other hand, KT15 and KT20 extracts generated ramifications in the SIVs basket, but the data collected did not reveal concentration dependence of this effect.

## Effects of six medicinal plant extracts on cartilage formation in zebrafish larvae

The early development of the head skeleton depends on many regulatory pathways, such as Wnt, Bmp, Hh signaling that are also involved in other developmental processes [50].

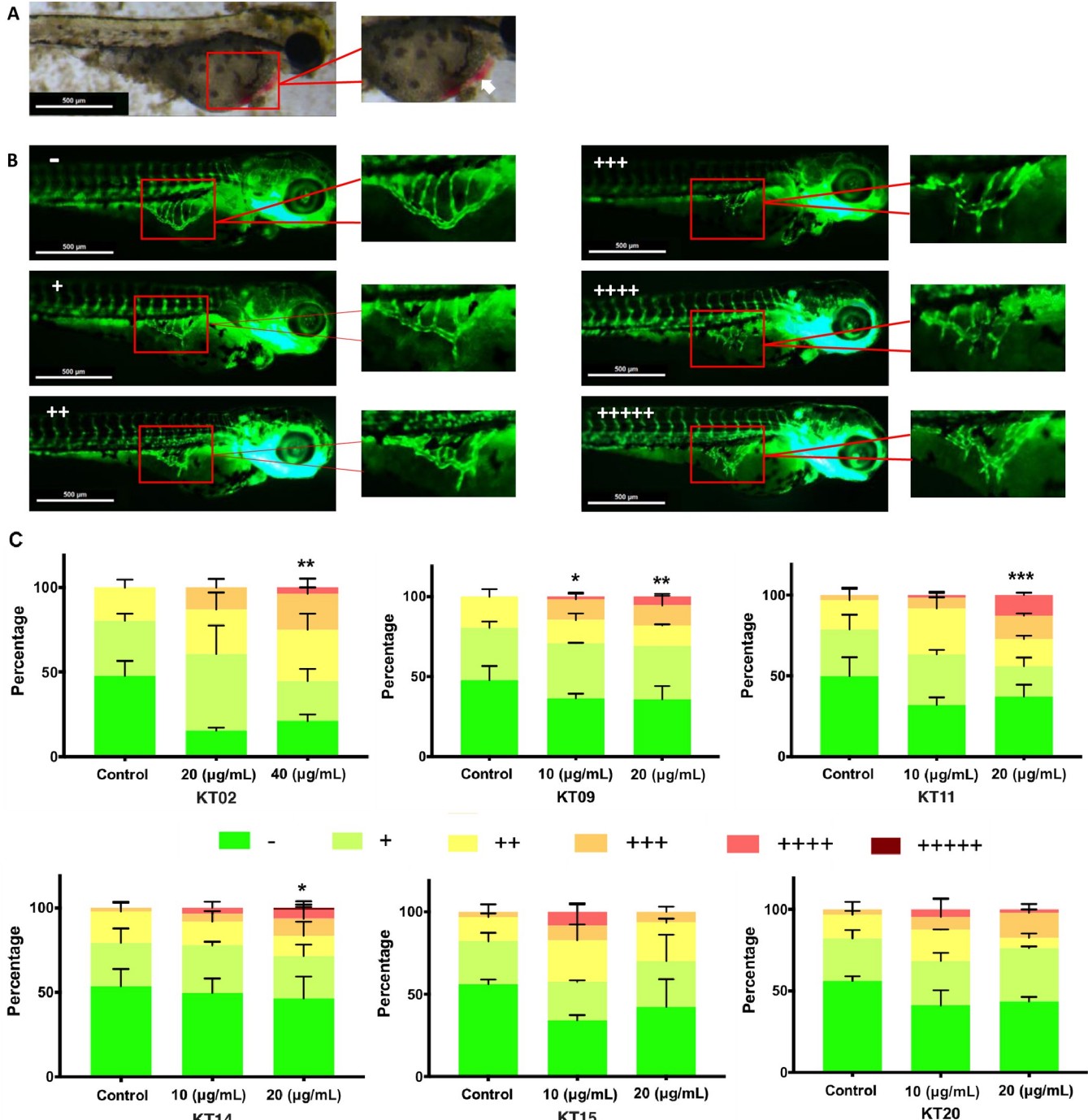

**Fig 3. The effects of six plant extracts on blood vessel formation in SIVs of *Tg (fli1: EGFP)* zebrafish larvae.** (A) Hemorrhage (white arrow) was observed in 3 dpf zebrafish larvae treated with KT14 150 μg/mL. (B) Microscopic view of blood vessel formation of zebrafish at 72 hpf in the absence and presence of ectopic angiogenesis of Sub intestinal vessels (SIVs), SIVs are marked by a red frame and shown at a higher magnification. The number of "+" is the number of sprouts projecting from the SIVs basket. (C) Quantitative analysis of the percentage of each angiogenic phenotype in SIVs after treatment with the different plant extracts. Sample groups were assessed according to "+", "++", "+++", "++++" and "+++++" representing the percentage of embryos with one, two, three, four or five sprouts projecting out of the SIV regions, respectively. Images were taken using ZEN software. Data present mean ± SD from three independent experiments. Statistical analysis was performed using a two-way ANOVA test, p-values < 0.05 *; 0.01 **; 0.001***.

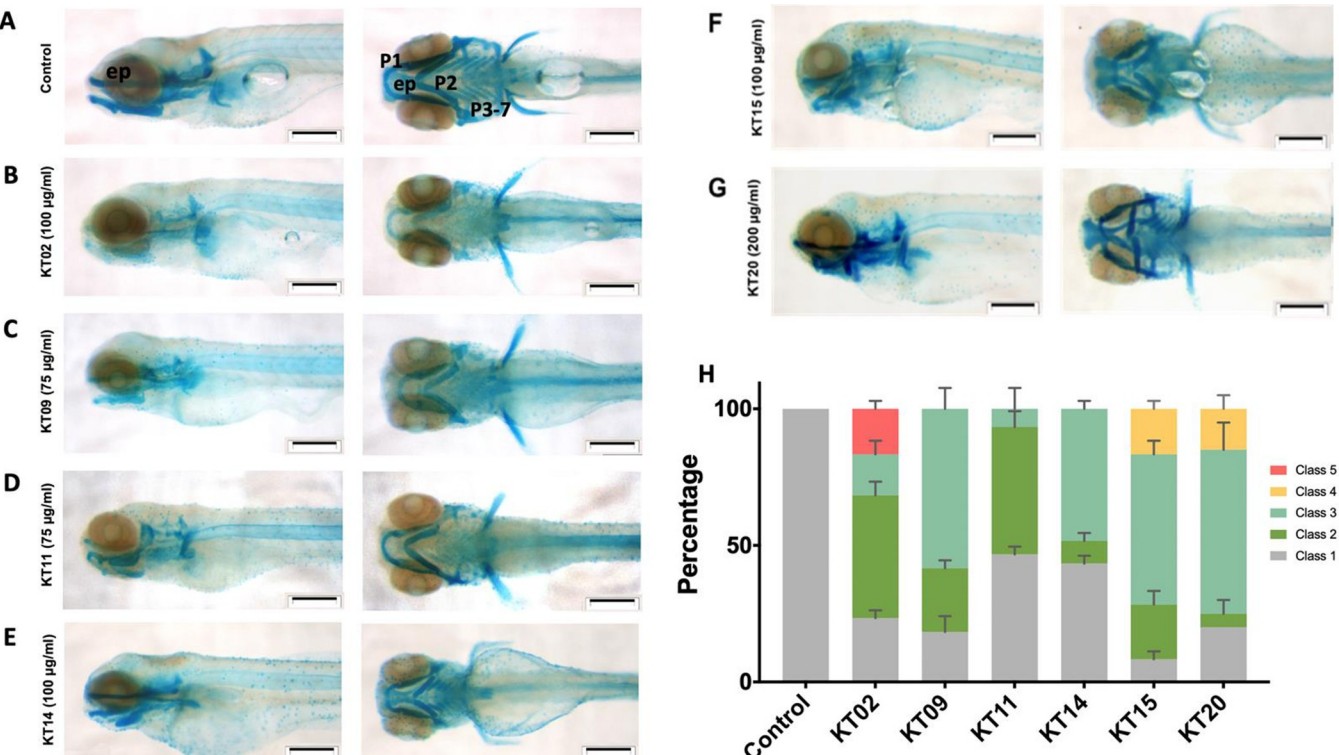

**Fig 4. Medicinal plant extracts exposure affects head cartilage formation in zebrafish larvae.** Whole-mount preparations of the larva expose to extract of KT02 (B), KT09 (C), KT11 (D), KT14 (E), KT15 (F) and KT20 (G). (A) Control. Scale bar: 200 μm. (H) Statistical analysis of jaw malformations as categorized into different classes. Class 1: No observable change (Control); Class 2: Weak staining in P1 to P7 region and ethmoid plate; Class 3: Shortened jaw structure (P1, P2 and ep); Class 4: Severe structural changes in all regions; and Class 5: strongly reduced or loss of all regions. Data from three independent replicates are presented as bar graph. Abbreviations: P1: Meckel's cartilage and palatoquadrate; P2: ceratohyal; P3-P7: ceratobranchial, and ep: ethmoid plate. Statistical analysis was performed using a two-way ANOVA test, p-values relative to control were always< 0.0001.

Malformations of the head are among the teratological effects induced by plant extracts. Therefore, we performed a more thorough analysis of effects on the head skeleton through enhanced visualization by staining of the cranial cartilage. We employed the Alcian blue dye, which can specifically combine with the cartilage matrix and thus visualize cartilage structures [51]. Whole-mount Alcian blue staining revealed that the head cartilage at 120 hpf had a different pattern in larvae treated with each plant extract at the concentration of $LC_{10}$ as compared to the control. The skull and pharyngeal cartilage structures in the control group had completely developed (Fig 4A), while in the treated groups, effects ranged from nearly absent (B) to lightly stained (C) cartilage, a shorter chondrocranium compared to the control group (B, C, F), to severely deformed cartilage elements (E, G) indicating that medicinal plant extracts hindered the differentiation of chondrocytes and thus induced decreases in chondrocyte number, cartilage matrix secretion and lead to morphological alterations (Fig 4A–4E).

Based on the staining observed of the pharyngeal arches (P1: Meckel's cartilage and palatoquadrate; P2: ceratohyal; P3-P7: ceratobranchial) and the ethmoid plate (ep) structure (Fig 4), we divided the deformities into 5 classes (Fig 4H). Class 1: No observable change (Control); Class 2: Mild staining in P1 to P7 region and ethmoid plate; Class 3: Shortened jaw structure (P1, P2 and ep); Class 4: Severe structural changes in all regions; and Class 5: strongly reduced or loss of all regions. Statistical analysis revealed that each extract's influence on zebrafish cranial cartilage formation was significant (p<0.0001), it increased from KT11, KT14, KT09,

KT02, KT20 to KT15 extracts. When exposed to KT11 and KT14 extracts, the percentage of larvae categorized as Class 1 (without any observable changes in Alcian blue staining) was still roughly 50% of the total larvae (Fig 4H), while it decreased to 8% upon KT15 exposure. Treatment with KT11, KT14, and KT09 only caused Class 2 and 3 malformations, whereas KT20 and KT15 generated significant proportions of severely malformed larvae (Class 4). For those larvae with shorter P1, P2, and ep (class 3), the head shape is clearly shorter (and more rounded) than the control at 120 hpf when viewed laterally (KT09 and KT14). Strong staining and extreme malformation of the chondrocranium, particularly the Meckel's and ceratohyals, were observed in larvae treated with KT20 (Fig 4G), while in contrast KT02-exposed larvae had a significantly higher proportion of larvae with nearly no Alcian blue staining compared to controls or to those exposed to other extracts.

## Effects of six medicinal plant extracts on locomotor activity of zebrafish larvae

To investigate possible effects of the plant extracts on zebrafish behavior, which might indicate modifications of neural development, we exposed zebrafish embryos/larvae to the plant extracts during 5 days before analyzing their motor behavior using an alternating 10 minutes dark and light protocol. In this type of test, it is known that zebrafish larvae are much more active in the dark, thus we observed a significantly increased time spent active, distance travelled and swimming speed during the dark phases for most of the conditions (Fig 5).

Upon exposure to plant extracts, we observed changes in swimming activity between the treated and control groups in both dark and light periods, as shown in Fig 5. Under the 10-min dark and 10-min light cycle alternate stimulation, 5 out of 6 plant extract-treated groups showed significant increases of swimming distance, time and speed in dark conditions compared to the control group, only the KT09 extract exposure significantly decreased the activity of larval zebrafish in the dark. The KT20 extract exposure group exhibits the largest increase in swimming behavior when compared to other extracts, in both the dark and light phases, followed by KT02. This observation suggests that these extracts may cause hyperactivity.

As mentioned above, the locomotor activity of the larvae in general was higher during the 10 min dark phase, compared to the light phase. An interesting exception is the group exposed to KT09, where the swimming distance, time, and speed did not differ dramatically between the two periods. We wondered whether this effect could be due to impaired vision in this group. However, when we look at the rapid transition moments from dark to light or the reverse transition from light to dark, respectively, we observe a "startle" peak especially in the swimming speed (Fig 5C). This peak is seen in control larvae, and in larvae treated with KT15 and KT09, indicating that these larvae do respond to, and thus perceive the change in lighting. The absence of such a peak in larvae treated with KT20 and KT02 may be due to their increased activity in the dark phase, while KT11- and KT14-exposed animals appear to be more tolerant to these surprising changes.

## Early developmental effects of the *Mahonia bealei* (KT09) extract

**Effect of *M. bealei* (KT09) extract on epiboly process.** Epiboly is characterized by spreading of the cell mass over the yolk sac—an important process that plays a prominent role in embryo gastrulation and specification of the dorsoventral axis [20, 52]. It is the first morphogenetic movement in zebrafish and amphibian embryos, leading to the expansion of blastoderm/ectoderm cells along the animal-vegetal axis, thereby covering the yolk cells and closing the blastopore. The early feature of dumbbell-shaped yolk at 8 and 10 dpf caused more

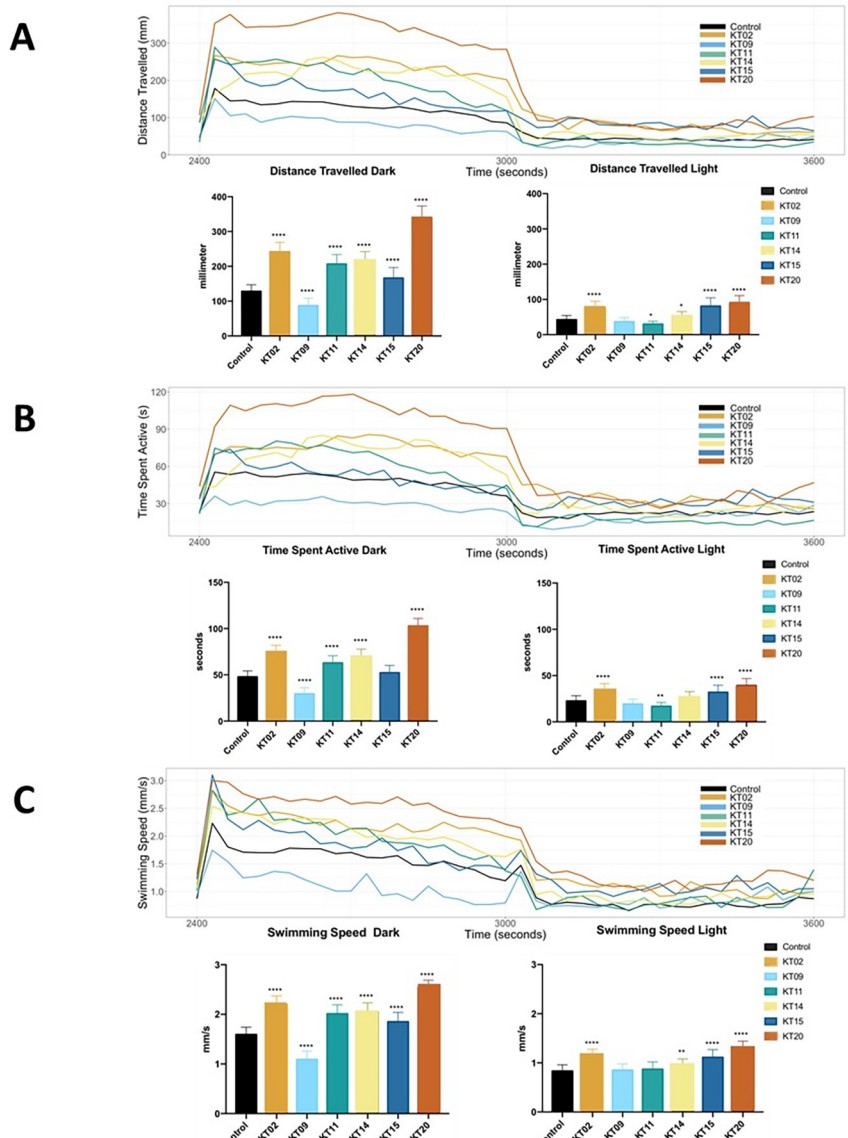

**Fig 5. Locomotor behavior of larval zebrafish following exposure to various plant extracts for 5 days.** The records of larval motion are shown in 30 second intervals over one cycle of a 10 min dark- 10 min light photoperiod. (A) Free swimming distance recorded during 10 min in the dark, followed by 10 min in the light. (B) Time spent active during 10-min intervals for each light state (dark or light). (C) Free swimming speed over 20 min of one dark- to-light cycle. Data are expressed as the mean ± SEM of three replicates (24 larvae per replicate). The asterisk represents a statistically significant difference when compared with the controls as described in Mat. and Meth.; * at $p < 0.05$, ** at $p < 0.01$ and **** at $p < 0.0001$ levels.

prominently by the KT09 extract could be best explained by a delay in the process of epiboly, where several layers of embryonic cells move towards the vegetal pole to finally nearly cover the entire egg. Epiboly depends on the dynamic regulation of the actin cytoskeleton, the process of extensive cell movements involves intracellular actin filaments and their associated myosin motor proteins [53]. Cortical F-actin is organized in the enveloping layer (EVL) cells at the start of blastodisc flattening. Later in epiboly, a punctate marginal actin ring appears ahead of the leading edge of external yolk syncytial layer (E-YSL), and F-actin is present in the vegetal cortex of the yolk cell [54]. We therefore decided to further characterize this defect by

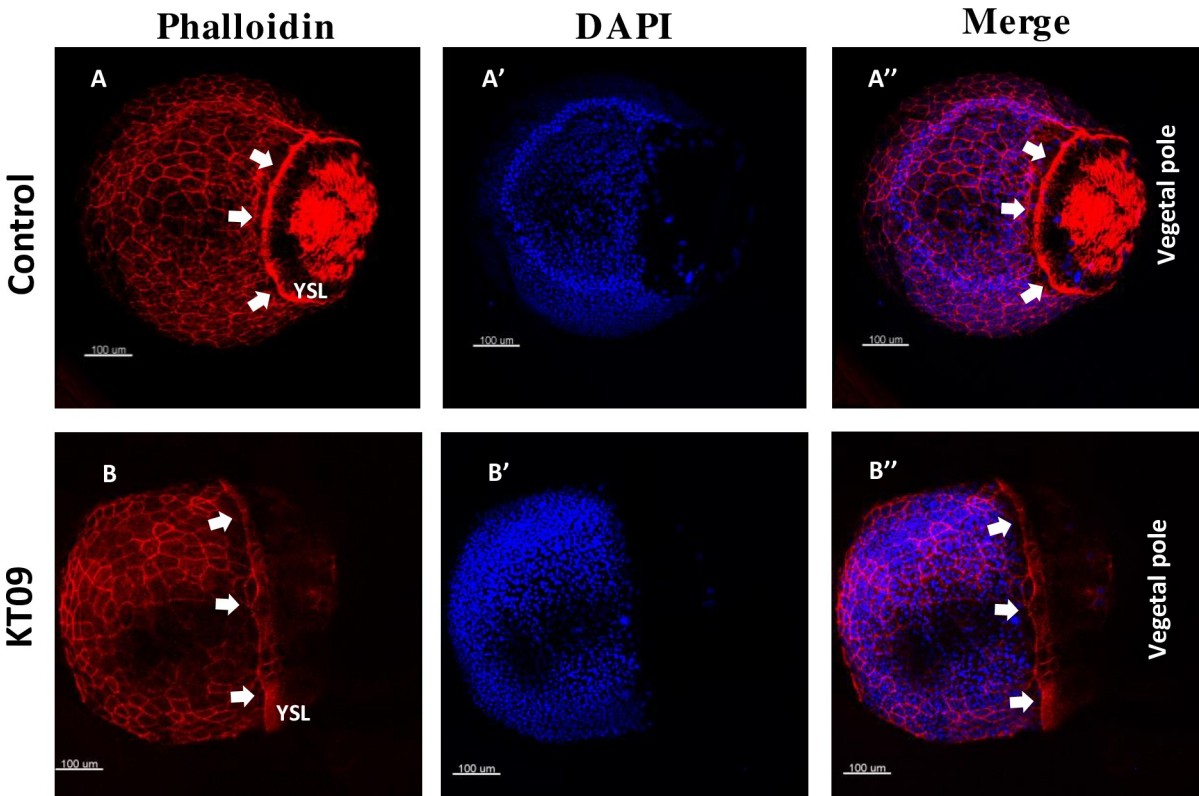

**Fig 6. Delayed epiboly progression and disruption of F-actin organization caused by KT09 extract treatment.** Embryos were treated with KT09 extract starting at 2hpf, while phalloidin and DAPI staining was performed at 80% epiboly (9hpf). (A-A") A control embryo at 90% epiboly stage, with a dense marginal actin ring (white arrows). (B-B") A representative embryo expose to KT09 extract shows epiboly retardation, and disruption of the EVL cortical F-actin, marginal actin ring and vegetal cortex F-actin. Scale bar: 100 μm.

following epiboly progression of blastoderm cells using phalloidin staining of filamentous actin (F-actin) fibers. At 9 hpf, when control embryos reached 90% epiboly, a regular and strong cortical F-actin was observed in EVL cells, while the vegetal cortex F-actin [55] formed bundles aligned along the animal-vegetal direction in the vegetal cortex of the yolk cell (Fig 6A). In addition, a strong F-actin ring was formed ahead of the leading edge of E-YSL (Fig 6A–6A"). KT09 extract (100 μg/mL) treatment resulted in a strongly reduced cortical F-actin in the EVL and a weak actin ring in a large majority of embryos. The organization of the vegetal cortex F-actin was also strongly disrupted, and epiboly progression was severely delayed (Fig 6B). These observations indicate the KT09 extract is causing a delay in epiboly by substantially impairing the actomyosin mechanisms in the yolk cell that drive this process.

**Effect of *M. bealei* (KT09) extract on cell migration.** After observing that the KT09 extract could delay epiboly cell movements in zebrafish embryos, we wondered whether a similar effect could be observed on mammalian cell migration. Mouse embryo fibroblast (MEF) cells were used to first investigate the effect of the KT09 effect on cell viability. The results revealed an IC50 value of 138.8 μg/mL, an IC10 of 4μg/mL and a NOEC of 2 μg/mL while no cells survived at 900 μg/mL of KT09 extract (Fig 7A and 7B).

To investigate the impact of the KT09 extract on cell migration *in vitro*, the ability of MEF cells to heal a wound applied to the cell monolayer was evaluated at the concentrations, respectively 0.4 μg/mL and 2 μg/mL. Wounds of comparable width and size were made in the wells of control and treated group. The cells in the control and 0.1% DMSO wells recovered at the

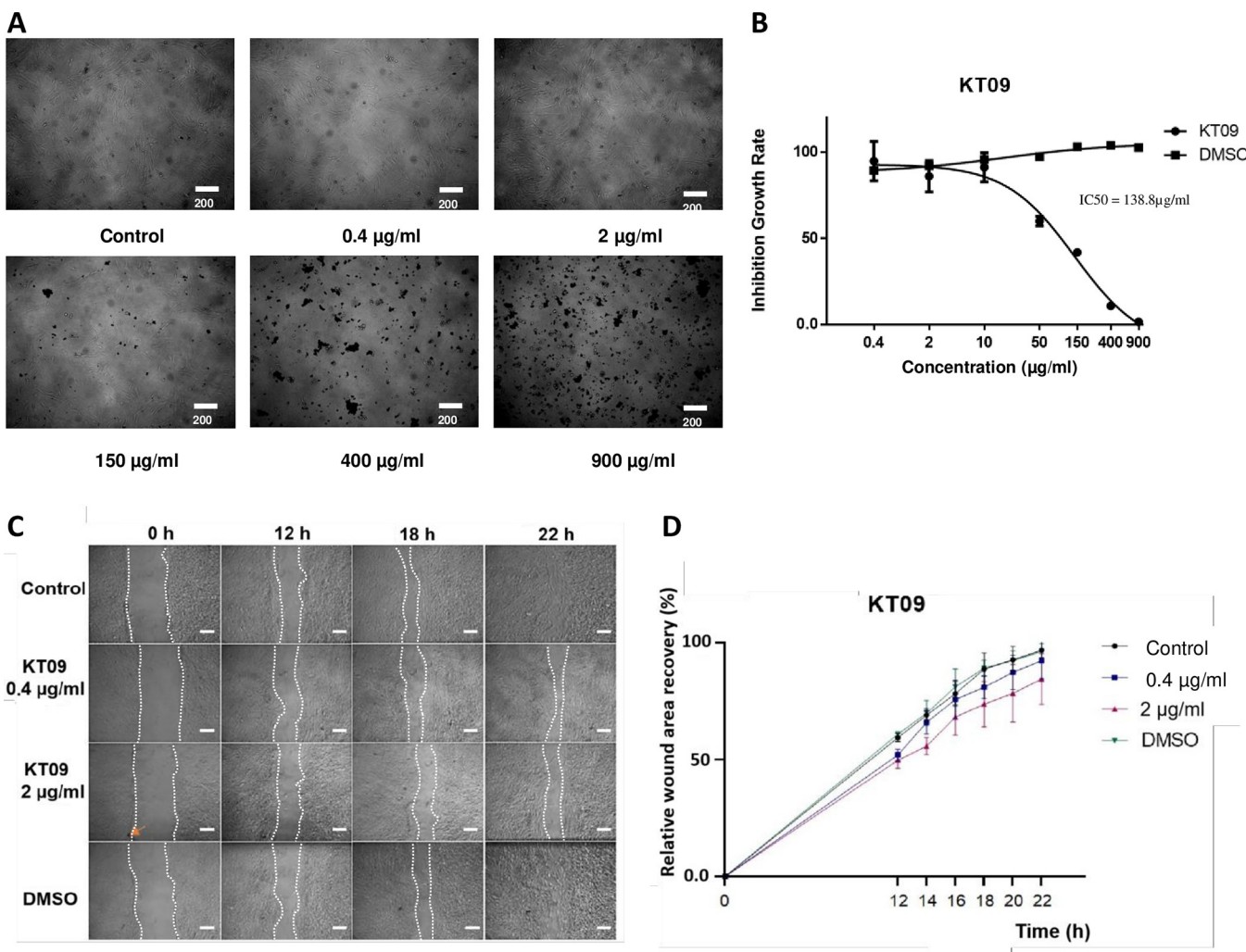

**Fig 7.** A. MEFs cell pictures after 48 hours of exposure to KT09 extract at different concentrations. B. Inhibition growth rate of MEFs cell exposed to KT09 extract in dose dependent manner ($IC_{50}$ = 138.8 µg/mL). C. Pictures of the scratch taken at four different time points 0, 12, 18, and 22 hours after being exposed to KT09 extract. White dotted lines indicate the cell boundaries. (KT09 0.4 µg/mL inhibits 0% cell proliferation; KT09 2 µg/mL inhibits 10% cell proliferation; orange arrow: extract residue; scale bar: 200 µm). D. Graph displaying the percentage of wound recovery following KT09 extract exposure at various time points.

same rate, indicating that the DMSO solvent content in the extract had no effect on the cells' wound healing abilities (Fig 7C). In contrast, the wounds recovered more slowly in cells exposed to KT09 extracts than in controls, with the percentage of wound area recovered in wells treated with 0.4 µg/mL and 2 µg/mL of KT09 being 93%±5% and 85%±5%, respectively, while the wound recovery rate in the control well was approximately 97% (Fig 7D). The result of the two-way ANOVA revealed that the difference in recovery between wells exposed to KT09 extract and control was statistically significant ($p<0.05$).

**Chemical components of the *M. bealei* (KT09) extract.** To gain some insight into the chemical compounds present in the *M. bealei* extract, we compared the extract to several reference alkaloids under the same HPLC-DAD conditions. The UV chromatograms at 270 nm revealed that the extract contained jatrorrhizine, isotetradine, 8-oxypalmatine, oxyberberine, 8-oxyberberubine, berberine, and palmatine (Fig 8, S2 and S3 Figs), with berberine and

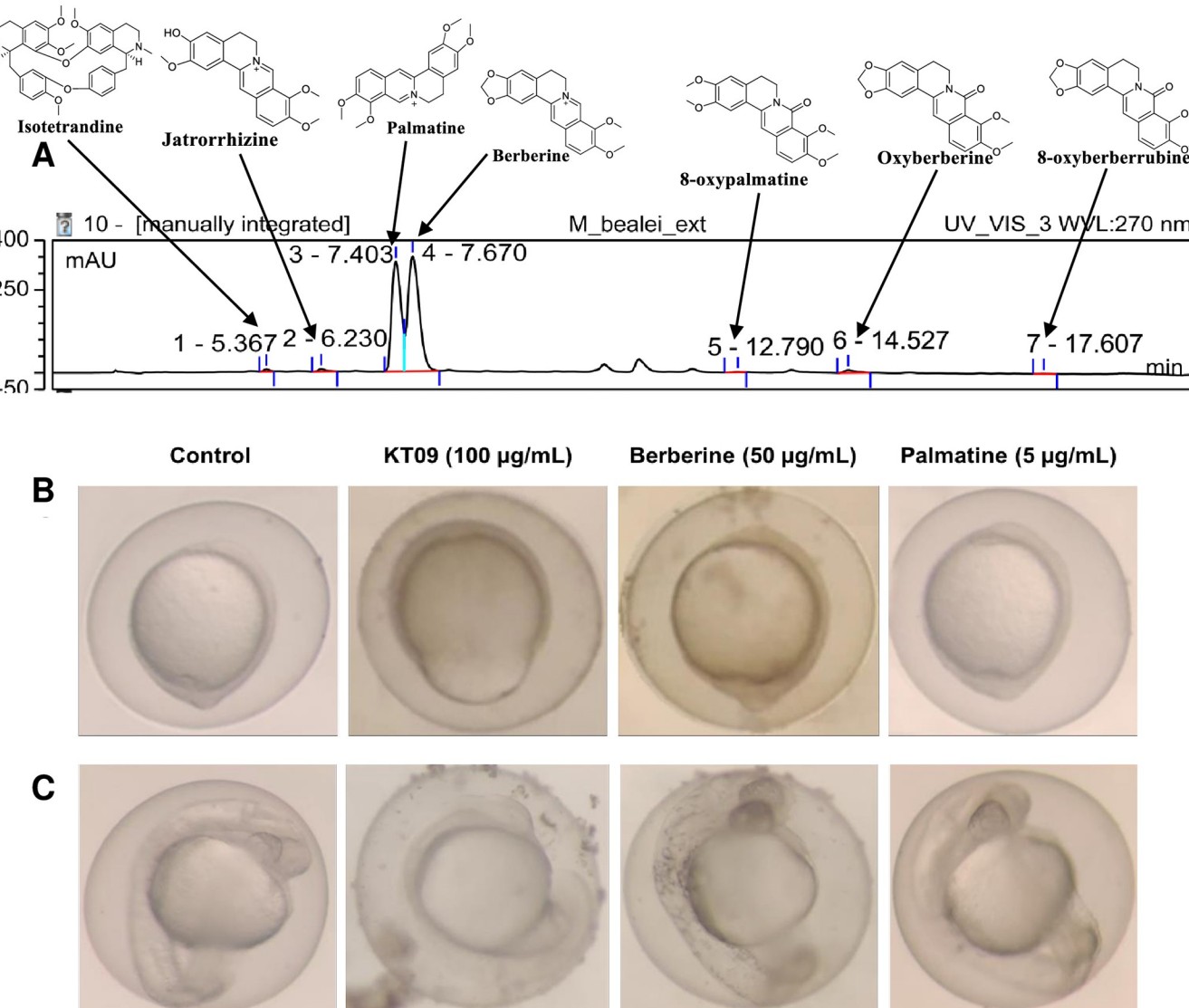

**Fig 8. Chemical composition of the *M. bealei* extract and activities of major compounds.** (A) HPLC chromatogram (at UV wavelength 270 nm) of the *M. bealei* extract is shown. The different detected compounds are indicated by arrows pointing at the corresponding peaks, with the two major peaks (peaks 3 and 4) corresponding to, respectively palmatine and berberine, while minor peaks (vertical ticks) correspond to jatrorrhizine, isotetradine, 8-oxypalmatine, oxyberberine, and 8-oxyberberrubine (B, C) Representative images of 10 hpf (B) and 24 hpf (C) zebrafish embryos exposed to *M. bealei* crude extract, berberine, or palmatine at the indicated concentrations.

palmatine contributing the vast majority of alkaloids [46] in the extract as shown by their largest peak area on the chromatogram (Fig 8A)(respectively 121 ± 9 mg/g and 15.2 ± 0.2 mg/g of the total extract) [46].

When we tested purified berberine and palmatine at the concentrations of 50 μg/mL and 5 μg/mL, higher than the respectively 12 μg/mL and 1.5 μg/ml in the KT09 extract, no effect was observed on epiboly retardation, vascular development, or general embryogenesis (Fig 8B and 8C).

**Effect of *M. bealei* (KT09) extract on gene expression in zebrafish embryos.** To further deepen our understanding of the effects caused by the plant extracts on zebrafish development, we decided to investigate their effects on whole genome gene expression using RNA-Seq

analysis. We decided to focus (first) on the earliest effect that we detected; the dumb-bell shape of the embryo observed during the epiboly process. We chose to treat the fertilized eggs with the KT09 extract at 100 μg/ml, as this was the treatment where this defect was most consistently observed, and we purified whole embryo mRNA from control and treated individuals at 8 hpf to perform transcriptomic analysis.

RNA-Seq analysis revealed the presence of 2855 differentially expressed genes (DEGs) (adjusted p-value < 0.05, fold-change > |1.5|), 1335 up-regulated and 1520 down-regulated genes (S1 Table) (GEO Series accession number GSE207770 on NCBI's Gene Expression Omnibus [56]).Among the genes whose expression was most highly affected (S2 Table), one striking observation is the presence of many transcription factor genes that are downregulated. These genes include several *hox* genes involved in anterior/posterior pattern specification and embryonic skeletal system morphogenesis [57], genes of the *pax* family instrumental in patterning and organ development, and the *nr2f5* gene coding for a nuclear receptor involved in upper jaw development [58]. Another strongly repressed gene is the *wnt8b* gene, which is involved in zebrafish neuron development and guidance [59, 60]. Among the most upregulated genes are *myo1g*, predicted to enable actin filament binding and motor activity, and *dachb* involved in pancreas and inner ear development [61, 62].

Gene ontology analysis was performed on the list of DEGs affected by the KT09 extract. We performed GSEA analysis, as it considers the extent to which a gene is up- or downregulated. This analysis identified many developmental processes to be affected, such as sensory and nervous system development, head development, cell differentiation, and pattern specification (S3 and S4 Tables). Main molecular functions affected were transcriptional regulation, but also receptor regulation and protein dimerization, as also identified by the biological processes of gene expression regulation and second messenger mediated signaling. Among the prominent signaling pathways affected, the most cited was the Wnt pathway, but also Shh and BMP signaling. A closer inspection using the Wikipathways database revealed that 13 genes in the canonical Wnt pathway were downregulated, while 13 genes of the noncanonical Wnt pathway were upregulated (Fig 9A and S5 Table). The *wnt8b* gene was among the most down-regulated genes, as well as the Wnt receptor genes *fzd1* and *fzd10*, while *wnt5b* was unaffected (Fig 9A). Note that the beta catenin gene *ctnnb2*, a central transcription factor for canonical Wnt signaling, was up-regulated 1.68-fold.

Focusing on the main defect observed in treated embryos, we collected a list of 42 genes whose mutation was shown to affect epiboly (zfin.org), among which 12 saw their expression changed by a factor of 1.5, and 19 genes displayed at least a 1.3-fold, but highly significant change in expression (Fig 9B and S5 Table). Other processes involving cellular movement were also significantly altered, such as cell migration, egg coat formation, neural crest cell migration (S3 and S4 Tables). Consistent with these observations, the actin cytoskeleton was detected as downregulated.

Several pathways were pointed out that may be involved in the defects in skeletal development that we observed, such as chondrocranium, cranial, and ceratohyal cartilage, as well as pharyngeal endoderm, BMP signaling, endochondral ossification, and collagen formation. Similarly, the alterations of movement behaviors could be related to DEGs connected to neural development and locomotion terms.

## Discussion

Traditional medicinal plants are widely, and increasingly used all over the world as an alternative or a complement to mainstream treatments. Their "soft" or "natural medicine" reputation is rooted in popular knowledge accumulated over generations of traditional healers, both

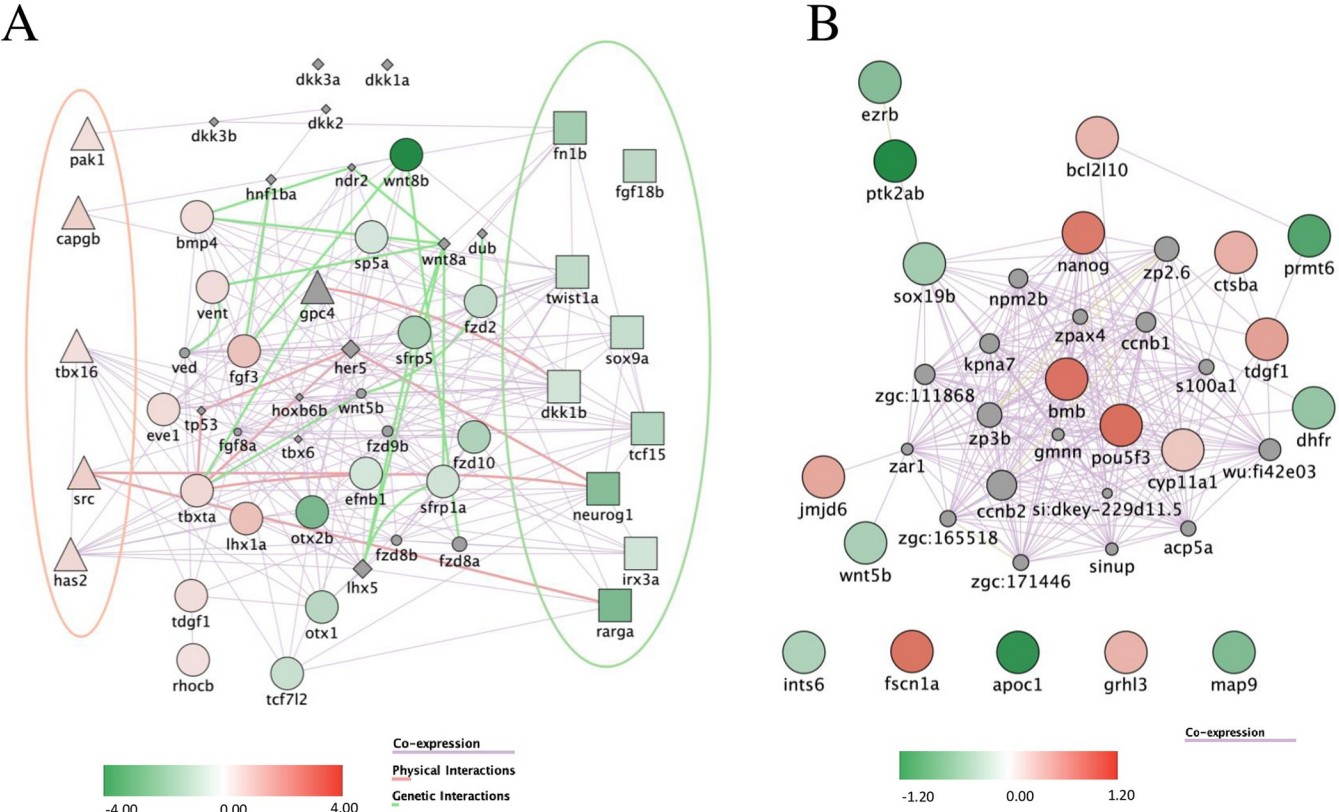

**Fig 9. Pathways affected upon treatment with *M. bealei* extract.** (A) Differentially expressed genes related to Wnt signaling upon KT09 treatment. A network was constructed based on the DEGs (p<0.05; fold-change > |1.4|). Shapes used indicate genes attributed to the canonical Wnt pathway (squares), the non-canonical pathway (triangles), both pathways (circles), or none (diamonds). The red and green ellipses highlight the genes that are specific to, respectively, the non-canonical and the canonical Wnt pathway. (B) Differentially expressed genes upon KT09 treatment related to epiboly. A network was constructed based on the DEGs (p<0.05; fold-change > |1.3|). The color intensity indicates the fold-change of down-regulated (green) and up-regulated (red) genes by KT09 treatment. The lines correspond to relations of co-expression (violet), physical interaction between encoded proteins (pink), and genetic interaction (light green).

concerning their pharmaceutical actions and their innocuity. However, in many cases, objective assessment of both their beneficial and their toxic potential is lacking, mainly due to the enormous workload and cost that these analyses would incur. Here, we set out to use the zebrafish as a cheap and easy model to test the effects of medicinal plant extracts on vertebrate development. Focusing on the embryogenesis, this approach allows us to evaluate effects on early development, but we also rapidly assess other pathways and processes that may play a role in pathogenesis at later life stages.

Traditional practitioners prepare the most common oral drugs by decocting in water for many hours at high temperature, and a few medicinal plants are soaked in alcohol at a concentration of around 40%. Both water and alcohol were used separately for extraction of all the plants we investigated, both because of their habitual uses and because they are typical solvents for polar and nonpolar chemicals. In the study's preliminary screening tests, aqueous extracts exhibited mostly no, or very minor activity. Therefore, we decided to restrict our study to ethanolic extracts of the relevant parts of the plants to ensure comparable extraction conditions for the different plants.

Looking at lethality and teratogenicity tests of all 12 extracts (Table 2), we observed that KT02 and KT14 caused lethality only at 72 and 96hpf, while extracts KT06, KT08, and KT19

**Table 4. Summary of the observed biological activities with active concentration.**

| Extract | Embryonic malformation | SIVs ramification | Cartilage formation | Locomotion Capacity |
|---|---|---|---|---|
| KT02 | Dumb-bell shaped yolk, Oedema, Necrosis (100 μg/mL) → death rate increased after 72hpf | Defect seen at >= 40 μg/mL | Weak staining of all regions (100 μg/mL) | Significant increases of swimming activity in the dark (100 μg/mL) |
| KT11 | Dumb-bell shaped yolk, Oedema, Necrosis, Curvature, Hatching delay (50 μg/mL) → death rate increased within 24hpf, and after 72hpf | Defect seen at >= 20 μg/mL | Weak staining of ceratobranchial and ethmoid plate (75 μg/mL) | Increased swimming activity in the dark (50 μg/mL) |
| KT14 | Dumb-bell shaped yolk, Oedema, Necrosis, Curvature, Embryos died once they hatched (100 μg/mL) →death rate increased after 48hpf | Defect seen at >= 20μg/mL | Shortened jaw structure (100 μg/mL) | Increases of swimming activity in the dark (100 μg/mL) |
| KT15 | Dumb-bell shaped yolk, Oedema, Necrosis, Hemorrhage, Curvature (50 μg/mL) → death rate increased after 72hpf | No effect | Shortened jaw and structural change (100 μg/mL) | Increases of swimming activity in the dark (100 μg/mL) |
| KT20 | Mild Dumb-bell shaped yolk, Oedema, Curvature (100 μg/mL) →death rate increased within 24hpf | No effect | Shortened jaw and structural change in all regions (200 μg/mL) | Largest increase in swimming behavior (200 μg/mL) |
| KT09 | Dumb-bell shaped yolk, Oedema, Necrosis, Curvature, Hatching delay (50 μg/mL), clearly developmental retardation → death rate increased within 24hpf | Defect seen at >= 10 μg/mL | Shortened jaw structure (75 μg/mL) | Decreased activity in the dark →swimming activity not different between dark and light periods (75 μg/mL) |
| | F–actin | Cell migration | | RNA seq |
| KT09 | reduced cortical F-actin in the EVL and weak actin ring, vegetal cortex F-actin disrupted →epiboly progression severely delayed (100 μg/mL) | MEF cell migration inhibited→ wound recovery slowed (2 μg/mL) | | Gene expression and pathways modified: Wnt signaling, epiboly, cell migration, egg coat formation, neural crest cell migration. . . (100 μg/mL) |

did not cause any developmental defects at 24 and 48hpf, but did so at later stages with similar $EC_{50}$ values as other extracts. One possible explanation is that the younger embryos may have been protected from the toxic or teratogenic compounds by the chorion. Future experiments, using dechorionated embryos or microinjecting the extracts into the chorion, will enable us to solve this issue.

Focusing on specific developmental defects at early stages, we selected 6 extracts to test their effects at non-lethal concentrations ($LC_0$ = NOEC (no observed effect concentration), $LC_{10}$, and $LC_{25}$), that we summarize in Table 4 for easier reference. One of the earliest defects observed to some extent with all extracts was the delayed epiboly, giving rise to dumb-bell shaped yolks at 8 and 10hpf. As explained above, epiboly is the first morphogenetic movement of cells in zebrafish that depends on the dynamic regulation of cortical F-actin (filamentous actin) in the EVL [52]. Focusing on the treatment with KT09, we confirmed a significant reduction of the F-actin signal and a clear delay in the epiboly progress at 9hpf (Fig 6). These F-actin networks are essential for the circumferential constriction of the margin during late epiboly stages, and for the structural integrity of the yolk cell [52, 54, 63]. Treating embryos with the actin-depolymerizing agent cytochalasin at 50% epiboly delayed epiboly, blocked blastopore closure, and led to yolk cell lysis [54], while inhibiting signaling through the small G protein Rac1 disrupted the F-actin organization and/or delayed epiboly [55]. RNA-Seq analysis confirmed that genes whose mutation affects epiboly [52, 63] (zfin.org) had their mRNA levels at 10hpf changed at least 1.3-fold upon treatment with KT09 (Fig 9B).

In this context, it is also striking that RNA-Seq analysis of the 10hpf transcriptome revealed that the Wnt pathway was most significantly affected upon KT09 treatment. Studies following single cell movements revealed that the inhibitor C59, disrupting Wnt protein maturation, causes delayed epiboly in zebrafish embryos [64]. Wnt5b was shown to be required for early cell movements by regulating Rac1, Cdc42, and finally actin filament dynamics [65], however

the *wnt5b* gene expression was unaffected. Our results indicate that the canonical Wnt pathway, induced by Wnt5b and acting through ß-catenin, is down-regulated, while the non-canonical pathway is up-regulated upon KT09 treatment. Note also that our analysis included maternal mRNA, mainly from the yolk, thus the observed changes may be due to transcriptional effects, but also differential maturation or degradation. Exactly how these changes may affect early epiboly, or other effects that we observed, and how far they may occur upon treatment with other extracts remains to be investigated.

Another striking effect of the various plant extracts is the hatching delay observed with KT09 and KT11 treatment. Although this delay could be caused by inhibition of the proteases He1a and Cathepsin L ([66] produced by the zebrafish hatching gland, it is also noteworthy that among the most strongly downregulated genes by KT09 at 10hpf are the *slc39a7* and *sp63* genes (S5 Table), both shown to be required for hatching gland development [67, 68].

It is tempting to speculate that defects in cell migration during epiboly around 8 to 10hpf may be related to the observed defects in SIV formation at later stages. However, as pointed out above, the concentrations causing SIV defects are much lower than those leading to delay of epiboly. SIV formation is mainly associated with vEGF signaling at later stages, it is however interesting to see that among the genes downregulated by KT09 at 10hpf, the *s1pr1* gene and the *hoxd4a* gene are both described to be required for SIV formation [69, 70].

At later stages (5dpf), we observed various deformities and defects in head cartilage formation, which may be caused by interference of the different extracts with various signaling pathways that are known to be involved in this complex process [50, 71]. In the case of KT09 treatment, the observed cartilage defects (weak staining, shortened chondrocranium) may be linked to the observed downregulation of Hox genes, known to control cranial cartilage patterning, however further studies would be required for this and other extracts. Similarly, the observed effects of the different extracts on locomotor behavior are probably linked to the observed perturbations in early central nervous system development.

In recent years, detection of anti-tumor activities of traditional medicinal plants attracted much attention, often based on testing the effects of extracts on proliferation, survival, or apoptosis of tumor cell lines [72–74]. Here, we observed for the first-time defects in epiboly and SIV formation following treatment with several of the extracts investigated, which led us to test whether one of them, the KT09 extract would also act on mammalian cell migration. We observed a significant inhibition of cell migration of primary mouse embryonic fibroblasts, without affecting their survival or proliferation. Recently, it was shown in a small-scale, pilot chemical screening experiment that 30% of the compounds causing disruption of epiboly/gastrulation in zebrafish embryos were also able to inhibit human cancer cell motility without affecting their proliferation. One of these drugs was shown to inhibit cell invasion by metastatic human cancer cells and cancer metastasis in mouse models [75]. Thus, our results indicate that some of the extracts tested here may find a new application in preventing cancer metastasis.

Chemical alkaloid composition of the KT09 crude extract was analyzed, revealing that the extract contained jatrorrhizine, isotetradine, 8-oxypalmatine, oxyberberine, 8-oxyberberubine, berberine, and palmatine. Quantification revealed that berberine and palmatine were the major compounds (respectively 12% and 1.5% w/w) in the *M. bealei* (KT09) extract [46]. Berberine was previously shown to be most probably responsible for the plant's pharmaceutical anti-fungal properties. However, preliminary findings examining the effects of berberine and palmatine on the epiboly process in zebrafish did not reveal a similar impact to that of crude extract. At 8–10 hpf, the dumb-bell shaped yolk phenomenon was not observed in embryos treated with concentrations of berberine or palmatine, even at a much higher dose than that present in the crude extract (Fig 8B and 8C). According to https://medlineplus.gov/druginfo/

natural/1126.html#Safety, berberine is safe to be used at doses of 1.5 grams daily for 6 months, with possible side effects on the digestive tract probably due to its anti-bacterial action on the intestinal microflora. Similarly, palmatine has antibacterial activity, while it was also shown to be toxic for human hepatocytes and cardiomyocytes *in vitro* [76], however none of these activities would be predicted to cause the developmental defects investigated here. Consequently, it still remains unclear which phytochemicals account for this observed effect. Another intriguing finding was that KT09 treatment had the highest impact on the Wnt pathway, which is critical in cardiomyocyte development. Indeed, previous reports have shown that *M. bealei* (KT09), and specifically berberine, can have beneficial effects on heart function [77–79]. For the perspective of this research, therefore, alternative applications of KT09 should be exploited and the chemical constituents responsible for its properties should be identified.

## Conclusions

Taken together, our results show that the zebrafish embryotoxicity test reveals potential toxic or developmental effects caused by extracts from traditional medicinal plants (Table 4). The traditional use of the different medicinal plants investigated here range from anti-inflammatory, stomachache, and duodenum ulcers (KT14), anti-tumor (KT09, KT11, KT14, KT20), antioxidant, antimicrobial to antiviral [72, 80, 81]. A literature review of their chemical composition revealed compounds of particular interest, such as quercetin, ursolic acid, berberine and palmatine [72, 82, 83]. These substances are found in more than one plant and have been reported to exhibit activities that are associated with those in our findings [76, 84–87]. Our results would advice caution concerning the toxicity of KT12, while the effects on zebrafish embryo development may suggest caution when using the extracts to treat young children or pregnant women. However, it is also obvious that further research will be required to assess the effects on human systems, the bioavailability of the compounds to the developing embryo, and the dose used for treatment. Further studies are also needed to confirm the mechanisms of toxico/pharmacological properties of these medicinal plants, as well as which components contribute to such activities. Studies using zebrafish larvae can help directing toxicological research towards specific pathways, but may also provide hints to identify new, beneficial properties that may finally lead to new applications or drugs.

## Supporting information

**S1 Fig. Survival curves for each experimental day upon treatment with 6 different medicinal plant extracts.**
(TIF)

**S2 Fig. HPLC chromatograms of *M. bealei* extract.** HPLC chromatograms of *M. bealei* extract and in-house reference compounds, including jatrorrhizine, isotetradine, 3-hydroxy-8-oxopalmatine, 8-oxypalmatine, oxyberberine, rugosinone, 8-oxyberberrubine, berberine, and palmatine.
(TIF)

**S3 Fig. Extended chromatograms of *M. bealei* extracts.** Extended chromatograms (-6 to 16 mAu) of *M. bealei* extract, jatrorrhizine, isotetradine, 3-hydroxy-8-oxopalmatine, 8-oxypalmatine, oxyberberine, rugosinone, and 8-oxyberberrubine. Matched compounds (jatrorrhizine, isotetradine, 8-oxypalmatine, oxyberberine, and 8-oxyberberrubine) are marked on the chromatogram of the extract.
(TIF)

**S4 Fig. Structures of the compounds detected in *M. bealei* extract.**
(TIF)

**S1 Table. Differentially expressed genes (DEGs) at 10hpf upon treatment with KT09.**
(XLSX)

**S2 Table. Genes most affected by KT09 treatment.**
(DOCX)

**S3 Table. GSEA-GO-term analysis of DEGs.**
(DOCX)

**S4 Table. All enrichment analyses.** ORA indicates "Over-Representation analysis" using UP-
or DOWN-regulated genes, GSEA indicates 'Gen Set Enrichment Analysis", used databases
were GO, KEGG, Panther, Reactome, Wikipathways, and a home-made zebrafish mutant
gene-phenotype list, "description" and "size" gives the name and size of the corresponding
term, "overlap" indicates the number of genes in the dataset, "enrichmentRatio" indicates the
enrichment factor in the term, while pValue and FDR (False Discovery Rate) indicate the sta-
tistical significance of the enrichment, "userId" holds the list of the overlapping genes in the
dataset.
(XLSX)

**S5 Table. Genes whose mutation affects selected processes.**
(XLSX)

## Acknowledgments

We would like to thank the GIGA zebrafish platform (H. Pendeville-Samain) for taking care of
and delivering the zebrafish larvae, the GIGA imaging platform for their help and support
with microscopy, the GIGA genomic platform for sequencing, and the GIGA bioinformatics
platform for data analysis.

## Author Contributions

**Conceptualization:** Hoang Giang Do, Thi Kim Thanh Nguyen, Hong Diep Le, Hai The
    Pham, Lai Thanh Nguyen, Marc Muller.

**Data curation:** Trung Kien Kieu, Lai Thanh Nguyen, Marc Muller.

**Formal analysis:** My Hanh Tran, Gustavo Guerrero-Limon, Renaud Nivelle, Hai The Pham.

**Funding acquisition:** Hai The Pham, Lai Thanh Nguyen, Marc Muller.

**Investigation:** My Hanh Tran, Thi Van Anh Nguyen, Hoang Giang Do, Trung Kien Kieu, Thi
    Kim Thanh Nguyen, Gustavo Guerrero-Limon, Laura Massoz, Renaud Nivelle, Jérémie
    Zappia.

**Methodology:** My Hanh Tran, Thi Van Anh Nguyen, Hoang Giang Do, Trung Kien Kieu, Thi
    Kim Thanh Nguyen, Gustavo Guerrero-Limon, Laura Massoz, Renaud Nivelle, Jérémie
    Zappia, Lai Thanh Nguyen, Marc Muller.

**Project administration:** Thi Kim Thanh Nguyen, Hong Diep Le, Lai Thanh Nguyen, Marc
    Muller.

**Resources:** Thi Kim Thanh Nguyen, Hong Diep Le.

**Supervision:** Hong Diep Le, Hai The Pham, Marc Muller.

**Validation:** My Hanh Tran, Hoang Giang Do, Hai The Pham, Lai Thanh Nguyen, Marc Muller.

**Visualization:** Gustavo Guerrero-Limon, Laura Massoz, Jérémie Zappia.

**Writing – original draft:** My Hanh Tran, Thi Kim Thanh Nguyen, Hai The Pham, Lai Thanh Nguyen, Marc Muller.

**Writing – review & editing:** Hai The Pham, Lai Thanh Nguyen, Marc Muller.

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
