## [Decision Letter · Decision Letter 0]

17 Aug 2023

PONE-D-23-05393Testing biological actions of medicinal plants from northern Vietnam on zebrafish embryos and larvae: developmental, behavioral, and putative therapeutical effects.PLOS ONE

Dear Dr. Muller,

Thank you for submitting your manuscript to PLOS ONE. After careful consideration, we feel that it has merit but does not fully meet PLOS ONE’s publication criteria as it currently stands. Therefore, we invite you to submit a revised version of the manuscript that addresses the points raised during the review process.

Both reviewers' have requested a number of clarifications and revisions, please address their comments point by point.  A completely agree with reviewer #1 that speculation about conclusions on the impacts of various compounts on human pregnancy are inappropriate and should be removed until relevant mammalian studies have been carried out.  

We look forward to receiving your revised manuscript.

Kind regards,

Michael Klymkowsky, Ph.D.

Academic Editor

PLOS ONE

Journal Requirements:

This research was funded by ARES (Académie de Recherche et d'Enseignement Supérieur)(https://www.ares-ac.be/fr/), grant number PRD17: "Exploring the medical, (eco)-toxicological and socio-economic potential of natural extracts from Northern Vietnam.” M.H.T. was a fellow from ARES, G.G-L. had a fellowship from EU project MSCA_ITN PROTECTED and M.M. is a "Maître de Recherche" at FNRS.

This research was funded by ARES (Académie de Recherche et d'Enseignement Supérieur), grant number PRD17: "Exploring the medical, (eco)-toxicological and socio-economic potential of natural extracts from Northern Vietnam.” M.H.T. was a fellow from ARES, G.G-L. had a fellowship from EU project MSCA_ITN PROTECTED and M.M. is a "Maître de Recherche" at FNRS.

This research was funded by ARES (Académie de Recherche et d'Enseignement Supérieur)(https://www.ares-ac.be/fr/), grant number PRD17: "Exploring the medical, (eco)-toxicological and socio-economic potential of natural extracts from Northern Vietnam.” M.H.T. was a fellow from ARES, G.G-L. had a fellowship from EU project MSCA_ITN PROTECTED and M.M. is a "Maître de Recherche" at FNRS.

Reviewers' comments:

Reviewer's Responses to Questions

**Comments to the Author**

1. Is the manuscript technically sound, and do the data support the conclusions?

Reviewer #1: Partly

Reviewer #2: Yes

2. Has the statistical analysis been performed appropriately and rigorously? 

Reviewer #1: Yes

Reviewer #2: Yes

3. Have the authors made all data underlying the findings in their manuscript fully available?

Reviewer #1: Yes

Reviewer #2: Yes

4. Is the manuscript presented in an intelligible fashion and written in standard English?

Reviewer #1: Yes

Reviewer #2: No

5. Review Comments to the Author

Reviewer #1: I found this paper interesting and the study design thorough. I have some concerns, that upon addressing, would improve this manuscript for publication

Introduction:

I disagree with the phrase ‘not considered as an animal’ (line 79). Zebrafish larvae are definitely still animals, however it is a larval stage organism and therefore not a protected species.

Methods:

Were any of the researchers blinded to the treatment groups when screening for phenotypes? Please state.

In the cell scratch assay why was DMSO not dosed at 0.5% like in the fish assays?

Did you deyolk the embryos for the RNA extraction? If yes, please state. If no, please consider the maternal RNA present in the yolk sac that influences development.

Data and analysis section states that experiments performed 3 times- please specify if this was technical replicates, or biological replicates – ie 3 separate clutches of eggs and in which case.

Discussion:

I think the discussion reads, in places, as more of a summary of results rather than a discussion of the findings, and the implications for the research field. I therefore think this manuscript would benefit from a summary table of all the effects observed by the drugs, collating the results of each assay. I suggest re-writing the discussion to limit the reiteration of the results, and to address the following questions:

Fig 8 reveals that the two main chemical components are not influencing the gross morphological phenotype. Can you discuss this further, and explain why you might anticipate these results?

The authors mention KT11 might have two different death causing processes, can you hypothesise why/what these might be in relation to the development of the zebrafish? Do you have plans to look at RNA at different time points to elucidate this? Why might it be important?

I am also concerned about the sweeping assumption that the outcomes of this study should be considered when treating pregnant women with these compounds. I don’t think that this is addressed in this study to any extent to make these claims, ie there are no human specific studies, no information about crossing the placenta, whether a prescribed therapeutic dose would reach the same toxic levels. I think making this assumption is sensationalist at this point, and therefore further research is required to base these claims. Perhaps there is more literature that can be discussed in this capacity here?

Figures:

Please add gridlines to Table 2. Can you also include the number data from Supplementary Table 1 here somehow, and include the numbers from the control groups. This will support your argument that these phenotypes are more prevalent in the treated groups, and are not standard developmental issues that are common such as cardiac oedema and spinal curvature.

Figure 3, please state the P values for each bar, what is the significance (*) determined by? What groups are being compare and by what statistical test?

Figure 4F is missing KT15 annotation

Add ‘light’ and ‘dark’ labels to graphs in Figure 5 for increased clarity.

Figure 6 – add control and KT09 labels to A and B

Fig 7B – X axis title is obscuring the labels. 7C – please define the cell boundary.

No lines visible on Figure 9

Reviewer #2: The manuscript “TESTING BIOLOGICAL ACTIONS OF MEDICINAL PLANTS FROM NORTHERN VIETNAM ON ZEBRAFISH EMBRYOS AND LARVAE: DEVELOPMENTAL, BEHAVIORAL, AND PUTATIVE THERAPEUTICAL EFFECTS.” aimed to evaluate the biological effect of plants used as Vietnamese traditional medicine on parameters of embryo development, in order to assess the possible toxic effects.

First of all, I would like to compliment the authors for the effort in investigating the toxic potential of plant extracts, which could represent a serious threat to humans. The analysis of biological potential and cellular effects of plant extracts is essential to comprehend and assure the use of these plants.

Overall, the context of the study was well introduced, and the results and discussion are well presented. However. This reviewer recommends language revision, especially on ‘Material and methods’ section. .

SPECIFIC COMMENTS:

Main text:

Standardize the units of time measurement (hours post-fertilization or days post-fertilization).

Line 460-461 - Rewrite the sentence for better understanding.

Describe with better clarity the results of visual behavior.

Line 550 – 551 - Rewrite the sentence for better understanding.

Very short conclusion. I suggest rewriting it to provide significant insights regarding the findings.

Figures

Figure 4 F – The identification of the group is missing.

Figure 5 – Standardize the colors of each extract tested.

Figure 8 – Rewrite the caption for better understanding.

6. PLOS authors have the option to publish the peer review history of their article (what does this mean?). If published, this will include your full peer review and any attached files.

Reviewer #1: **Yes: **Siobhan Crilly

Reviewer #2: No

---

## [Author Response · Author response to Decision Letter 0]

11 Sep 2023

Reviewer #1: I found this paper interesting and the study design thorough. I have some concerns, that upon addressing, would improve this manuscript for publication.

Introduction:

I disagree with the phrase ‘not considered as an animal’ (line 79). Zebrafish larvae are definitely still animals, however it is a larval stage organism and therefore not a protected species.

We agree that this was an overstatement and changed the text: "the zebrafish embryo up to the free-feeding stage at 5 days post fertilization is generally considered as larval stage; hence it can be used without raising ethical issues." (line 80 in the mark-up version).

Methods:

Were any of the researchers blinded to the treatment groups when screening for phenotypes? Please state.

Strictly speaking, the researchers were not blinded to the treatment groups, however "each treatment group was analyzed without preconception, especially regarding the specific defect observed." Our hope was indeed to not see too many defects. 

We added additional explanation in the new manuscript (line 161-163)

In the cell scratch assay why was DMSO not dosed at 0.5% like in the fish assays?

It is known that zebrafish larvae will not suffer from 0.5% DMSO, which is actually what we also see, as we always compare treated and non-treated. We also wanted to be able to at least test higher concentrations of the extracts. For the scratch assay, lower concentrations were used, so we preferred to lower the DMSO concentration here.

Did you deyolk the embryos for the RNA extraction? If yes, please state. If no, please consider the maternal RNA present in the yolk sac that influences development.

Thanks for this question: indeed, we were interested in the differential mRNA pool upon treatment, thus we included the yolk. We mention this now in the Mat. and Meth. section and in the discussion. It does not affect our main conclusions.

"We chose to use whole larvae, including the yolk, as we intended to determine the entire RNA pool for evaluation of transcription, but also maturation and degradation." (lines 262-263).

"Note also that our analysis included maternal mRNA, mainly from the yolk, thus the observed changes may be due to transcriptional effects, but also differential maturation or degradation." (lines 805-807).

Data and analysis section states that experiments performed 3 times- please specify if this was technical replicates, or biological replicates – ie 3 separate clutches of eggs and in which case.

Thanks for the comment. It was always biological replicates: All experiments were performed at least three times, mostly on different dates but always on different clutches from different parents (lines 311-313).

Discussion:

I think the discussion reads, in places, as more of a summary of results rather than a discussion of the findings, and the implications for the research field. I therefore think this manuscript would benefit from a summary table of all the effects observed by the drugs, collating the results of each assay. I suggest re-writing the discussion to limit the reiteration of the results, and to address the following questions:

We actually really tried to avoid reiteration in the discussion, we now tried to do even better, and we include a summary table as suggested (Table 4, line 658). Thanks for that suggestion. However, consider that sometimes repetition is required to integrate a specific discussion point.

Specifically, we moved the lethality and teratogenicity discussion of all 12 extracts to the "Results" section (lines 364-370), leaving only the proposal that the chorion may protect younger embryos in the discussion (lines 746-751). We also shortened the epiboly discussion (lines 661-667), as that was indeed redundant to the text in the "Results"

Fig 8 reveals that the two main chemical components are not influencing the gross morphological phenotype. Can you discuss this further, and explain why you might anticipate these results?

Thanks for the question. We added text to the discussion, which may lead to anticipate our results, however we preferred to verify, and it is what we found.

"According to https://medlineplus.gov/druginfo/natural/1126.html#Safety, berberine is safe to be used at doses of 1.5 grams daily for 6 months, with possible side effects on the digestive tract probably due to its anti-bacterial action on the intestinal microflora. Similarly, palmatine has antibacterial activity, while it was also shown to be toxic for human hepatocytes and cardiomyocytes in vitro [78], however none of these activities would be predicted to cause the developmental defects investigated here." (lines 909-914)

The authors mention KT11 might have two different death causing processes, can you hypothesize why/what these might be in relation to the development of the zebrafish? Do you have plans to look at RNA at different time points to elucidate this? Why might it be important?

Thanks for the comment. Now the sentences were changed to " The survival rate of embryos was reduced at 24hpf, constant until 72hpf, and reduced again at 96hpf (S1 Fig), implying that this extract causes death at an early stage (before 24hpf) and again later, possibly due to the hatching delay. RNA-Seq analysis at different time points would help to understand the mechanisms involved." (lines 435-438)

I am also concerned about the sweeping assumption that the outcomes of this study should be considered when treating pregnant women with these compounds. I don’t think that this is addressed in this study to any extent to make these claims, ie there are no human specific studies, no information about crossing the placenta, whether a prescribed therapeutic dose would reach the same toxic levels. I think making this assumption is sensationalist at this point, and therefore further research is required to base these claims. Perhaps there is more literature that can be discussed in this capacity here?

Indeed, we probably insisted too heavily on this assumption, although it is a reason to use zebrafish larval development as a test. We now adress this point specifically in the discussion. Note that for example the use of berberine, present in many extracts, is not recommended for children and pregnant women, according to: 

(https://medlineplus.gov/druginfo/natural/1126.html#Safety).

" Our results would advice caution concerning the toxicity of KT12, while the effects on zebrafish embryo development may suggest caution when using the extracts to treat young children or pregnant women. However, it is also obvious that further research will be required to assess the effects on human systems, the bioavailability of the compounds to the developing embryo, and the dose used for treatment. Further studies are also needed to confirm the mechanisms of toxico/pharmacological properties of these medicinal plants, as well as which components contribute to such activities. Studies using zebrafish larvae can help directing toxicological research towards specific pathways, but may also provide hints to identify new, beneficial properties that may finally lead to new applications or drugs." (lines 987-995)

We actually think that a lengthy discussion of this issue would be out of scope for this manuscript.

Figures:

Please add gridlines to Table 2. Can you also include the number data from Supplementary Table 1 here somehow, and include the numbers from the control groups. This will support your argument that these phenotypes are more prevalent in the treated groups, and are not standard developmental issues that are common such as cardiac oedema and spinal curvature.

Thanks for this suggestion. We now integrated former table S1 into the text, as Table 3, as this seemed the most feasible option. In addition, we added a column for controls to this table.

Figure 3, please state the P values for each bar, what is the significance (*) determined by? What groups are being compare and by what statistical test?

Comparison is always against controls (untreated). The statistical tests used are now indicated in the legends of Figures (lines 478-479 for Fig 3 and lines 536-537 for Fig 4)

Figure 4F is missing KT15 annotation

Sorry for the mistake. We addressed it. 

Add ‘light’ and ‘dark’ labels to graphs in Figure 5 for increased clarity.

Thank you for your suggestion. We made the changes.

Figure 6 – add control and KT09 labels to A and B

Thank you for your comment. The labels were already added.

Fig 7B – X axis title is obscuring the labels. 7C – please define the cell boundary.

We are sorry, but we don’t get the point for Fig 7B. All labels are visible on all versions. For Fig 7C, we added white dotted lines to indicate the cell boundary in the shown examples.

No lines visible on Figure 9

The lines were indeed not very visible. We hope that it is better now, while we also try not to make it too confusing.

Reviewer #2: The manuscript “TESTING BIOLOGICAL ACTIONS OF MEDICINAL PLANTS FROM NORTHERN VIETNAM ON ZEBRAFISH EMBRYOS AND LARVAE: DEVELOPMENTAL, BEHAVIORAL, AND PUTATIVE THERAPEUTICAL EFFECTS.” aimed to evaluate the biological effect of plants used as Vietnamese traditional medicine on parameters of embryo development, in order to assess the possible toxic effects.

First of all, I would like to compliment the authors for the effort in investigating the toxic potential of plant extracts, which could represent a serious threat to humans. The analysis of biological potential and cellular effects of plant extracts is essential to comprehend and assure the use of these plants. Overall, the context of the study was well introduced, and the results and discussion are well presented. However, this reviewer recommends language revision, especially on ‘Material and methods’ section.

Thanks for the positive evaluation. The language was reassessed thoroughly

SPECIFIC COMMENTS 

Main text: 

Standardize the units of time measurement (hours post-fertilization or days post-fertilization).

We agree, we switched to hpf in the entire manuscript.

Line 460-461 - Rewrite the sentence for better understanding. Describe with better clarity the results of visual behavior.

We admit that this was somewhat misleading, by mentioning the control larvae, it is actually the case for almost all larvae. "In this type of test, it is known that zebrafish larvae are much more active in the dark, thus we observed a significantly increased time spent active, distance travelled and swimming speed during the dark phases for most of the conditions (lines 543-545)

Line 550 – 551 - Rewrite the sentence for better understanding.

This was indeed misleading. We now changed it to "When we tested purified berberine and palmatine at the concentrations of 50 µg/mL and 5 µg/mL, higher than the respectively 12 µg/mL and 1.5 µg/ml in the KT09 extract, no effect was observed on epiboly retardation, vascular development, or general embryogenesis (Fig 8B and 8C)." (lines 641-643)

Very short conclusion. I suggest rewriting it to provide significant insights regarding the findings.

Thank you for your comment. The conclusion now has been changed and a summary table (Table 4) was added.

Figures

Figure 4 F – The identification of the group is missing.

Sorry for this. It was added.

Figure 5 – Standardize the colors of each extract tested.

Thank you for the comment. The colors are now exactly matched for each extract.

Figure 8 – Rewrite the caption for better understanding.

There was a mistake in the B and C explanation. We hope the caption is now clearer " (A) HPLC chromatogram (at UV wavelength 270 nm) of the M. bealei extract is shown. The different detected compounds are indicated by arrows pointing at the corresponding peaks, with the two major peaks (peaks 3 and 4) corresponding to, respectively palmatine and berberine, while minor peaks (vertical ticks) correspond to jatrorrhizine, isotetradine, , 8-oxypalmatine, oxyberberine, and 8-oxyberberrubine (B, C) Representative images of 10 hpf (B) and 24 hpf (C) zebrafish embryos exposed to M. bealei crude extract, berberine, or palmatine at the indicated concentrations." (lines 646-650).

---

## [Decision Letter · Decision Letter 1]

25 Oct 2023

Testing biological actions of medicinal plants from northern Vietnam on zebrafish embryos and larvae: developmental, behavioral, and putative therapeutical effects.

PONE-D-23-05393R1

Dear Dr. Muller,

We’re pleased to inform you that your manuscript has been judged scientifically suitable for publication and will be formally accepted for publication once it meets all outstanding technical requirements.

Kind regards,

Michael Klymkowsky, Ph.D.

Academic Editor

PLOS ONE

Additional Editor Comments (optional):

Reviewers' comments:

Reviewer's Responses to Questions

**Comments to the Author**

1. If the authors have adequately addressed your comments raised in a previous round of review and you feel that this manuscript is now acceptable for publication, you may indicate that here to bypass the “Comments to the Author” section, enter your conflict of interest statement in the “Confidential to Editor” section, and submit your "Accept" recommendation.

Reviewer #1: (No Response)

Reviewer #2: All comments have been addressed

2. Is the manuscript technically sound, and do the data support the conclusions?

Reviewer #1: Yes

Reviewer #2: Yes

3. Has the statistical analysis been performed appropriately and rigorously? 

Reviewer #1: Yes

Reviewer #2: Yes

4. Have the authors made all data underlying the findings in their manuscript fully available?

Reviewer #1: Yes

Reviewer #2: Yes

5. Is the manuscript presented in an intelligible fashion and written in standard English?

Reviewer #1: Yes

Reviewer #2: Yes

6. Review Comments to the Author

Reviewer #1: My thanks to the authors for carefully addressing all reviewer comments. The Discussion is much improved and I appreciate the effort made to answer my suggested questions.

My feedback is minor

1. The introduction needs proofreading – please see attached pdf for my suggestions

2. In the version I can see, the ‘concentration’ title in figure 7B obstructs the numbers on the X axis. This may be addressed later in your proofs.

Reviewer #2: (No Response)

7. PLOS authors have the option to publish the peer review history of their article (what does this mean?). If published, this will include your full peer review and any attached files.

Reviewer #1: **Yes: **Siobhan Crilly

Reviewer #2: No

---

## [Editor Report · Acceptance letter]

30 Oct 2023

PONE-D-23-05393R1 

Testing biological actions of medicinal plants from northern Vietnam on zebrafish embryos and larvae: developmental, behavioral, and putative therapeutical effects. 

Dear Dr. Muller:

I'm pleased to inform you that your manuscript has been deemed suitable for publication in PLOS ONE. Congratulations! Your manuscript is now with our production department. 

Kind regards, 

on behalf of

Dr. Michael Klymkowsky 

Academic Editor

PLOS ONE